# Valuing conservation and natural wealth: The blue economy of manta ray watching in the Maldives

Hannah M. Moloney[1,2,3]*, Maria I. Garcia Rojas[1&], Nina Rothe[1&],
Asia O. Armstrong[4,5], Kirsty Ballard[1,6], Florence Barraud[1,3], Farah Hamdan[7],
Anthony J. Richardson[8,9], Enas Mohamed Riyad[10], Tamaryn J. Sawers[1,3],
Kathy A. Townsend[2], Guy M. W. Stevens[1,3]

1 The Manta Trust, Catemwood House, Norwood Lane, Corscombe, Dorset, United Kingdom, 2 University of the Sunshine Coast, School of Science, Technology and Engineering, Hervey Bay, Queensland, Australia, 3 Maldives Manta Conservation Programme, M. Kureli, Buruzu Magu, Maafannu, Malé, Republic of Maldives, 4 IUCN SSC Shark Specialist Group, Dubai, United Arab Emirates, 5 University of the Sunshine Coast, School of Science, Technology and Engineering, Sunshine Coast, Queensland, Australia, 6 Coastal Oregon Marine Experiment Station, Oregon State University, Newport, Oregon, United States, 7 Queen's University Belfast, School of Biological Sciences, Northern Ireland, United Kingdom, 8 School of the Environment, The University of Queensland, St Lucia, Queensland, Australia, 9 Commonwealth Scientific and Industrial Research Organisation (CSIRO) Environment, BioSciences Precinct (QBP), St Lucia, Queensland, Australia, 10 Environmental Regulatory Authority, Malé, Republic of Maldives

---These authors contributed equally to this work.
* hannah.moloney@mantatrust.org

## Abstract

Amid declining manta ray populations globally, the well-established and growing manta ray tourism industries generate substantial economic benefits and aid protective legislation for these threatened elasmobranchs. As flagship species, manta rays are a drawcard for marine wildlife tourism and a gateway for engaging the public and communities in conservation. Healthy marine ecosystems are the key drivers of employment and economic sustainability for island nations such as the Maldives. However, there are many stakeholders competing for these shared resources, which can result in environmental degradation. Economic valuations are a powerful tool for justifying the conservation efforts of threatened species and natural areas, especially in light of competing stakeholders. Using tour operator surveys ($n = 106$) and data mining, this study provides an updated assessment of manta ray watching tourism in the Maldives and represents the first national valuation of its direct economic and socio-economic benefits. In 2021, manta ray tourism in the Maldives generated an estimated US$227.3 million, including US$39 million on manta ray focused diving and snorkelling excursions, and US$188.3 million in related tourist expenditure, representing 2.6% of the national Gross Domestic Product. This industry appears to have grown around 380% since 2008 (US$8.1 million) and manta ray watching is now offered by 80% of tourism operators nation-wide. Our findings revealed that

---

**Data availability statement:** The data underlying the results presented in the study are available through Zenodo DOI 10.5281/zenodo.18252206.

**Funding:** HMM was funded by the University of the Sunshine Coast Research Training Program scholarship. The funders had no role in the study design, data collection and analysis, decision to publish, or preparation of the manuscript.

**Competing interests:** NO authors have competing interests.

manta rays hold intrinsic value and cultural significance within local communities. Acknowledging this, the flow-on benefits to the community extend beyond this industry, reaching local businesses, employed staff, and the government with the direct economic benefits of the manta ray tourism industry are estimated at over US$311 million per year. Such value highlights the significance of manta rays for the Maldives and the need for effective management centred on manta ray conservation to safeguard future prosperity and mitigate the potential impact of tourism on manta ray populations.

## Introduction

Wildlife tourism can support both economic growth and the conservation of natural resources and ecosystems [1–4]. Categorised as a cultural ecosystem service, wildlife viewing provides essential non-material benefits, including recreational opportunities, aesthetic and spiritual experiences, learning and cultural enrichment [5]. These services are predicated on a shared valuation of the intrinsic beauty of wild species, which motivates tourists to visit and incentivises local communities to protect these natural assets [5]. However, policymakers are frequently challenged by the need to balance the requirements of multiple stakeholders competing for limited environmental resources [6]. Highlighting the socio-economic profitability of a well-managed wildlife tourism industry can help strengthen the case for protecting threatened species and ecologically important habitats by demonstrating their tangible value compared to extractive industries [7–9]. In successful frameworks, visiting tourists benefit from enhanced wellbeing and education [5,10], while local communities gain economic stability through employment and diversified income streams [9], which can encourage greater conservation efforts by both operators and visitors. Ultimately, these interactions create a powerful incentive for governments to establish robust protection measures and secure long-term stakeholder support [11,12].

Ongoing management and monitoring of protected natural resources represent a significant financial commitment for governments, encompassing staff salaries, governance structures, and physical facilities [6,13]. To maximize conservation outcomes within constrained budgets, focusing efforts on charismatic species has emerged as a globally effective strategy for biodiversity conservation [14,15]. The selection of a species to serve as a conservation lead must be well-balanced and consider not only the intrinsic and ecological aspects (e.g., risk of extinction), but also the cultural, social, and economic context of the region where the species will be promoted [16,17]. As such, marine tourism focusing on a charismatic species can make important contributions to biodiversity conservation and provide an "umbrella" of protection for lesser-known species [15]. This sector has expanded rapidly since the 1980s [1,11,18,19], generating direct revenue for operators and indirect financial flows for governments and local businesses [4,20–22]. In remote locations with few alternatives, well-managed megafauna tourism provides a critical pathway for communities to utilise their natural resources sustainably [19,23].

Shark and ray watching is a rapidly growing area of marine wildlife tourism, with activities established in at least 42 countries, primarily within low- to middle-income nations, including island nations [3,8,24,25]. Economic valuations consistently demonstrate that the tourism value of focal species can far exceed their value in extractive fisheries [3,24,26]. For instance, Anderson & Ahmed [27] in 1993 estimated the lifetime value of a single live shark in the Republic of Maldives was US$33,500, contrasted with US$32 for a landed shark. Despite the economic potential, coral-reef associated sharks and rays are among the most vulnerable marine taxa; nearly 59% of these species are threatened with extinction according to the International Union for Conservation of Nature (IUCN) Red List of Threatened Species, driven largely by overfishing and exacerbated by habitat degradation and climate change [28,29].

In regions where tourism infrastructure exists, shark and ray tourism offers a viable alternative to extractive practices. Currently, 29 countries have established shark and ray sanctuaries, which typically prohibit the commercial fishing, trade, possession, or sale of these taxa within their Exclusive Economic Zones [30–32]. These sanctuaries are often located in small-island nations across the Pacific Ocean, Caribbean Sea, and Indian Ocean [32], where reliance on marine resources is high [33] but enforcement capacity may be limited [34]. Recognising the economic importance of these species, the Maldives' government implemented a national shark sanctuary in 2010 [35,36]. While rays and skates were first protected from the export trade in 1995/6, comprehensive legislation introduced in 2014 made it illegal to catch, kill, or harm all rays and skates, effectively making the Maldives a combined shark and ray sanctuary [36]. This legislative shift ended a century-long targeted shark (*miyaru* in the Maldivian Dhivehi language) fishery that utilised gillnets and longlines to harvest juvenile and adult individuals from 27 species for fins, meat, skin, liver oil, and jaws [37]. Unlike specialised offshore fisheries in other regions that target pelagic species in open water [38], the Maldivian shark fishery was a localised, multi-species operation that directly exploited coastal and reef-associated habitats shared by the burgeoning tourism industry [27,37]. These fisheries historically targeted many species that are now central to the Maldives' tourism appeal, including whale sharks (*Rhincodon typus*), whitetip reef sharks (*Triaenodon obesus*), blacktip reef sharks (*Carcharhinus melanopterus*), grey reef sharks (*Carcharhinus amblyrhynchos*), scalloped hammerhead sharks (*Sphyrna lewini*), and tiger sharks (*Galeocerdo cuvier*) [27,37,39,40].

## Manta ray ecology and tourism

Manta rays (*en madi* in Dhivehi) are zooplanktivorous giants found throughout tropical and subtropical oceans [41]. They aggregate seasonally at predictable locations that facilitate socialising and key life history functions including feeding, courtship and mating, predator avoidance, cleaning, and thermoregulation [42]. The predictable and reliable nature of these aggregations makes manta ray watching (MRW) a highly sought-after tourist activity [43–45] involving recreational diving, snorkelling, and boat-based observations of these animals in the wild [3]. The high value of these encounters is particularly evident in the Maldives, where tourists are willing to pay more for manta ray encounters than for those with sea turtles and reef sharks [46]. By 2013, the global MRW industry was valued at US$140 million per year (i.e., tour operator revenue and tourist expenses), across >25 countries, with the Maldives identified as the world's leading destination with 101 MRW sites and 157,000 annual dives and snorkels [3].

The Maldives has successfully leveraged its marine biodiversity as a primary draw for international visitors since 1972 [47,48]. Tourist arrivals have grown consistently, reaching over 1.3 million in 2021 [49,50]. Tourism is the backbone of the Maldives economy, accounting for over a quarter of the nation's US$8.9 billion Gross Domestic Product (GDP) in 2021 [49], with ~12% of the total labour force employed in resorts in 2022 [51]. Manta ray and shark watching are especially important components of the dive tourism industry [39,44,52].

However, this heavy economic dependence makes the nation acutely vulnerable to the climate crisis, habitat degradation, and overfishing [48]. While tourist arrival trends are often influenced by a complex interplay of political stability, global commodity prices, and natural disasters, the Maldives' 'brand' is fundamentally built on its idyllic natural environment and rich biodiversity of marine life [48]. Consequently, any declines in the population health of charismatic marine species such

as sharks and rays poses a direct threat to national economic stability [9,18,39,44,53]. These threats are compounded by historical precedents of resource depletion; for example, unsustainable shark fishing during the early 1990s led to declining shark numbers at popular dive sites, resulting in documented economic losses in the dive-tourism industry [9,43]. Notably, unlike sharks, manta rays have never been commercially targeted by Maldivian fisheries, providing a unique baseline for studying a relatively 'pristine' population [54].

The 26 atolls of the Maldives support the world's largest known population of reef manta rays (*Mobula alfredi*), estimated to be 3,500 [55,56]. Mark recapture data of ~100,000 sightings recorded since 1989, and consistently since 2003, has identified over 6,200 unique individuals, though models suggest mortality rates need to be accounted for [55]. Additionally, the Maldives hosts a population of oceanic manta rays (*M. birostris*; at least for some of the year), with estimates suggesting an abundance of at least one order of magnitude greater than the ~1,000 individuals currently identified [57]. The reef manta ray population aggregates at 48 known key aggregation sites (>100 sightings) [58], most notably the Hanifaru Marine Protected Area (MPA) in Baa Atoll, which serves as a critical foraging site [58]. Their distribution is influenced by the South Asian Monsoon; *M. alfredi* individuals typically migrate across the atolls as the seasons change, resulting in seasonal site use, but year-round sightings within the country [44,54,58]. Conversely, *M. birostris* are only encountered consistently in the south of the Maldives at Gnaviyani Atoll (referred to as Fuvahmulah hereafter) during the NE Monsoon [57].

In 2008, the MRW industry in the Maldives was valued at US$15.5 million annually [3,44], though this figure was limited to direct revenue and did not account for broader socio-economic flow-ons, such as employee wages, government taxation, and secondary supply-chain benefits. Given the substantial growth in tourism over the last decade, the contemporary scale of this reliance remains largely unquantified. Both species of manta rays, *M. birostris* and *M. alfredi,* are globally threatened (listed as Endangered and Vulnerable respectively on the IUCN Red List) due to declining population trends and their conservative life history traits, including late maturation and low fecundity [59–65]. Their recovery capacity is among the slowest of all elasmobranchs [60,66], making them highly susceptible to anthropogenic pressures [42,67,68]. Even within protected waters, poorly regulated tourism can lead to habitat destruction, behavioural disruption, and sub-lethal injuries [36,69–71]. Consequently, accurately estimating the socio-economic and intrinsic value of these species is vital for balancing the interests of tourism, fisheries, and coastal development [36].

In this study, we present the first national estimate of the direct economic benefits of the MRW tourism industry in the Maldives. We extend upon previous valuations (i.e., 1997, 2011 and 2013 [3,44,46]) by utilising surveys and data mining targeting tour operators, tourists, and government agencies. We aim to: (1) quantify the economic benefits across the business, government, and community sectors; (2) evaluate the size, scope, impact, and management of the industry; (3) estimate the lifetime economic value of individual *M. alfredi* and *M. birostris*; and (4) analyse the intrinsic value of manta rays to local communities. We hope that these findings will emphasise the national importance of an industry built upon charismatic species, providing the empirical evidence needed to support the continuation of ray sanctuaries, greater investment in MPA management, and manta ray conservation.

## Methods

### Study design

Data on the revenue of MRW tourism in the Maldives in 2021 were obtained between 2022–2023 from survey responses and online data mining of tour operators and government reports. MRW recreational activities include either snorkelling or scuba diving with the specific intent of viewing *M. birostris* and *M. alfredi* in the wild, in places where manta rays frequent seasonally, periodically, or year-round, consistent with the criteria used in other studies [3,44,58] (see S1 Table for definitions). It can also be watching manta rays from a vessel; however, this has not been quantified in this study. Viewing manta rays that were encountered opportunistically, out of season, or not at a MRW site, were not included as they do not fit the conservative approach of this study as described by Anderson et al. [44]. MRW guests are those who participated in a MRW trip, which may include individuals who participated in multiple trips (S1 Table). A MRW trip is a single dive or snorkel session on a vessel. For example, a two-tank dive is accounted for as two separate trips.

The study was undertaken in the Republic of Maldives, a small island nation in the central Indian Ocean spanning 870 km in length (north to south) and 128 km across (east to west) at its widest (*3.2°N 73.3°E*) (Fig 1). There are 26 geographical atolls with distinct reef systems [72], grouped into 21 administrative regions (17 atoll regions and four cities – including Malé City, the capital) (Fig 1) [36,73]. The administrative regions organise the country for governance [73]. For this study, we have grouped the data by administrative regions, with the exception of Malé City which is amalgamated with Kaafu,

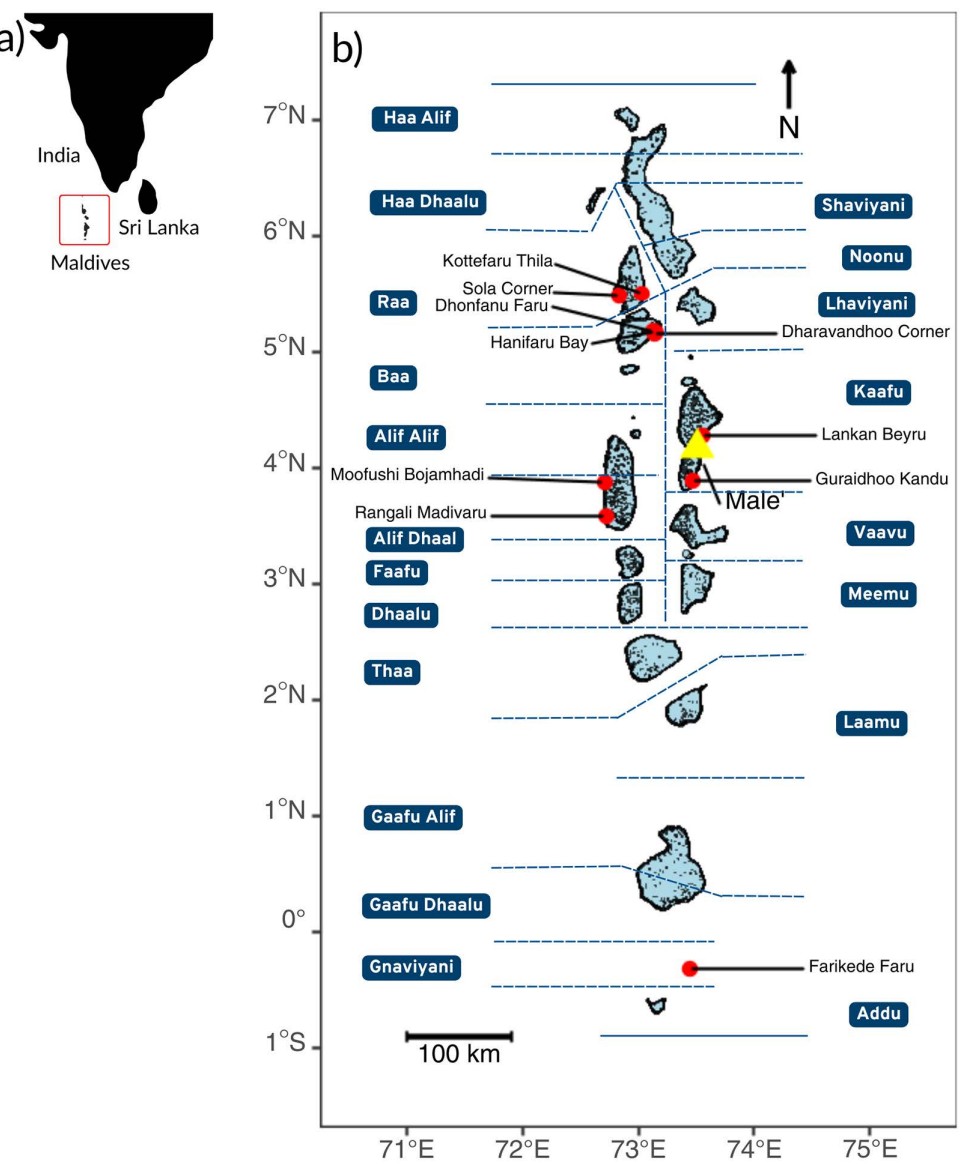

**Fig 1. Geographical distribution of the 10 primary manta ray watching sites (shown by red circles) in the Republic of Maldives'.** The 26 geographical atolls of the Maldives' archipelago are shown by the red box in panel a. The capital, Malé City is identified by the yellow triangle and the nation's 20 administrative divisions are highlighted in blue in panel b. Although Malé City is an administrative division, for this study we have grouped it with Kaafu, thus, only 20 divisions are shown. The manta ray watching sites correspond with those outlined in Table 2. This map was created in R Studio using reef features from Millennium Coral Reef Mapping Project (MCRMP; https://habitats.oceanplus.org); MCRMP validated maps provided by the Institute for Marine Remote Sensing, University of South Florida (IMaRS/USF) and Institut de Recherche pour le Développement (IRD, Centre de Nouméa), with support from NASA. IRD does not endorse these products [75].

to create more similarly sized areas. Thus, for the purpose of this study, the data is organised into 20 administrative regions that are used hereafter (Fig 1). Of the 1,192 islands in the Maldives, 193 are inhabited community islands (i.e., cities, towns, villages, fishing and farming communities with permanent human habitation), a further 172 are occupied with tourist resorts, and the remainder are protected islands, deserted islands, or are used for agricultural or industry purposes [49]. The population of the Maldives (n = 515,114) is centrally located with 41% of residents residing in Malé City (n = 211,908), with a further 46% on other community islands (n = 236,911), 3% on industrial islands (n = 13,831), and 10% on resort islands (n = 52,482) [74].

Tourism is concentrated in the regions surrounding the Velana International Airport (next to Malé City, Kaafu) but increasingly exists across all administrative regions [52,77]. In 2021, there were 354 registered tourism bases, such as resorts, liveaboards and community islands. Within these tourism bases, there were 538 registered tour operators, including dive and excursion activity centres (i.e., excursion centres offer recreational activities including snorkelling trips), and liveaboard vessels [78].

In this study, operators are independently run by third-party companies with separate teams, offices, schedules and vessels. For example, Four Seasons Resort Landaa Giraavaru, a 5-star resort island "base", has been divided into two "tour operators" as it has both a dive centre and an excursion centre, operated separately. Tour operators are classified into one of the following categories: resorts (n = 332) that are on private resort islands (n = 166), local activity centres (n = 107) that are on community islands (n = 39), and liveaboards (n = 99) that are vessels (n = 99). Thus, tour operators on resorts and community islands are referred to as land-based experiences and liveaboards as boat-based experiences. Because of the stark difference in prices of accommodation and other tourist expenses (TE), private resort islands were categorised as either "5-star luxury" (n = 48) or "regular" (n = 118) based on the star rating assigned online [79]. Tour operators were further classified according to whether they provided snorkelling or diving activities. Liveaboard cruises offer both snorkelling and diving activities on their trips, but given that most guests onboard are divers, they were considered dive-focused trips and were categorised as a diving centre only.

Private liveaboards are vessels hired privately for multiple-day trips are priced between US$12,175–84,000, but because of these high prices, it was not possible to decipher the price per dive and thus price per MRW dive. Private liveaboards, non-operational or closed bases (n = 43), were excluded from this study. There were also an additional 613 registered guesthouses [78], however the vast majority did not function as tour operators and outsource snorkelling or diving activities to other businesses on community islands (e.g., dive and excursion centres), and thus were excluded from this study. Hereafter, "tour operators" will include only dive and excursion centres based on resorts, community islands and liveaboard vessels.

## Tour operator surveys

To obtain information about the diving industry and manta ray tourism for the year 2021, a survey of tour operators was conducted, adapted from O'Malley et al. [3]. Prior to data collection, the survey was first piloted on experts in the industry with feedback incorporated into the survey design (n = 10, 100% response rate). The final online survey was distributed to all tour operators registered with the Ministry of Tourism (n = 538) [78] using Google Forms [80] between January – June 2022.

Two survey versions were designed, one for land-based operators (i.e., activity centres in resorts and on community islands) and one for boat-based operators (i.e., liveaboards). The survey consisted of a mix of open- and closed-ended questions, including submitter and tour operator details (questions n = 11), number of tourists and activity prices for MRW activities (questions n = 9), manta ray sites and sightings (questions n = 5), and the socio-economic and intrinsic value of MRW tourism to tour operators, guests and the community (questions n = 12) (S1 Appendix).

All 538 tour operators in the Maldives were contacted and offered the chance to voluntarily participate in the survey, with 106 operators responding (20% response rate). To estimate the economic benefit of the entire MRW

tourism industry in the Maldives and include data on all tour operators, we complemented the survey data with online data mining. Therefore, when registered tour operators did not complete the survey, or when survey data was missing, a standardised assessment was used, which involved data mining the websites and social media platforms of tour operators and travel agencies to collect available MRW tourism data (e.g., activity price, season length and marketing; S2 Table). If there was still missing data for MRW season length, it was sourced from the Manta Trust's *Maldives Manta Conservation Programme* long-term monitoring database [55] and from Harris et al. [58]. If values could not be sourced, the mean value was calculated by grouping tour operators based on their assigned activity type (i.e., snorkel, dive) and tour operator type (i.e., 5-star resort, resort, local activity centre, liveaboard) from both the survey and mined data.

To verify results from survey responses (i.e., activity price and season length), additional data mining was undertaken online for all participating tour operators. Surveys with major discrepancies between operator survey responses and internet data were excluded. For example, if the survey reported an activity price >20% different to that of the same company stated online, the value from the survey was disregarded. Obvious outliers in survey responses were excluded from this study in the initial data cleaning process ($n = 2$). The research was carried out under the *Environmental Protection Agency* Protected Species Research Permit (*EPA/2021/PSR-M09*) and in accordance with the University of the Sunshine Coast Human Ethics exemption (*E24002*).

## Economic benefits of manta ray watching tourism

The direct economic benefit from MRW tourism ($DEB_{MRW}$, US\$; Equation 1) in the Maldives in 2021 was estimated as the sum of: i) the tour operator revenue (*TOR*) from manta-ray-specific snorkelling ($TOR_S$, US\$), diving ($TOR_D$, US\$), and diving from a liveaboard ($TOR_{LAB}$, US\$) activities; ii) the revenue from tourist expenses (*TE*, US\$) including accommodation, food and beverages; iii) the revenue for government TAX (*TX*, US\$) and staff service charge (*SSC*, US\$) (proportions of *TOR + TE*); iv) staff salary revenue (*SSR*, US\$) for those who work directly with MRW tourism; and v) the Hanifaru MPA revenue (*MPAR*, US\$). Currency values throughout this study are presented as US\$.

$$DEB_{MRW} = TOR_s + TOR_D + TOR_{LAB} + TE + TX + SSC + SSR + MPAR \tag{1}$$

$TOR_S$ and $TOR_D$ (US\$) were calculated as the product of the activity price (*AP*, US\$), the number of guests per trip (*NG*, unitless), the number of trips per week (*NT*, weeks-1), and the number of weeks during which the activities were conducted/season length in 2021 (*NW*, weeks-1; equation 2a). Importantly, the revenue generated from tourist visitor entry fees ($EF_V$, US\$) to Hanifaru MPA (*MPAR*) was deducted from equation 2a and remained as revenue generated through MPA management (Equation 4).

$$TOR_S \text{ or } TOR_D = (AP \times NG \times NT \times NW) - (EF_V) \tag{2a}$$

$TOR_{LAB}$ (US\$) was calculated as the product of the value of the activity price (*AP*, US\$), the number of opportunities to snorkel and dive with manta rays (*NO*, unitless), the number of guests per trip (*NG*, unitless), and the number of trips per year in 2021 (*NT*, year-1; equation 2b). As $TOR_{LAB}$ already accounted for tourist expenses (*TE*) as they are sold as a package, we estimated that only half of the price per trip (50%) was used to cover diving expenses and thus was used to calculate the activity price (*AP*) using the overall number of dives offered per liveaboard trip. We then used the activity price (*AP*) and the number of opportunities (*NO*) to snorkel and dive with manta rays to calculate the price per MRW dive. Where values for the activity price (*AP*), the number of guests (*NG*), the number of trips (*NT*), and the number of opportunities (*NO*) were missing from tour operators (e.g., non-responding operators), a mean value was assigned.

$$TOR_{LAB} = AP \times NO \times NG \times NT \tag{2b}$$

To calculate the combined revenue (*TOR + TE*, US$), which is made up of tour operator revenue (*TOR*) and tourist expenses (*TE*), we determined the base-specific cost benefit transfer ratios using the activity price (*TE:AP*, US$), consistent with other studies [3,18,52,81]. These ratios were calculated by comparing the online price of 10 randomly selected bases (*TE*; including accommodation, food and beverages) with the price of the activity (*AP*; MRW snorkelling or diving). To calculate the tourist expense (*TE*), the mean price of accommodation per day was used at US$1,733 for 5-star resorts, US$628 for resorts, and US$116 for guesthouses on community islands, combined with a conservative estimate of food and beverage per day at US$150 for 5-star resorts, US$100 for resorts and US$50 on community islands [82]. Given that the price per day on a liveaboard includes food, beverages and accommodation, we assumed that the tourist expenses (*TE*) were 50% of the price per day (for each MRW dive, we calculated the tourist expenses (*TE*) using 50% of the price per guest on a liveaboard trip/nights per trip), calculated as US$189 for liveaboards. The cost benefit transfer ratios used were 13.265 for 5-star resorts, 7.233 for regular resorts, 2.703 for community islands and 2.586 for liveaboards. For example, if a MRW snorkelling trip price US$100 (*AP*) at a 5-star resort, there would be an additional US$1,227 in tourist expenses (*TE*) for accommodation, food and beverages, with the combined revenue (*TOR + TE*) calculating to US$1,327. Whereas for the same activity price of US$100 (*AP*) on a community island, there would be an additional US$159 in tourist expenses (*TE*), and the combined revenue (*TOR + TE*) would come to US$259. Given that the data mining exercise for tourist expenses was attempted in 2024, three-years after the surveys were distributed, the cost benefit transfer ratios relied on back-calculating the price of accommodation in 2021 by using the 2024 prices and adjusting for inflation of 12.6% [83]. Tourist expenses are not inclusive of international and domestic travel expenses, despite most of the visitors arriving from overseas via plane and having to travel to their accommodation [49].

The government corporate tax (*TX*, US$) and the staff service charges (*SSC*, US$) in the Maldives in 2021 were 25% of the value of the services provided (i.e., 15% and 10%, respectively) [84]. Advertised prices in the Maldives are subject to both the tax (*TX*) and the staff service charge (*SSC*) [52,84]. In this study, the revenue from tax (*TX*) and the staff service charge (*SSC*) was calculated for both the tour operator revenue and the tourist expenses (*TOR + TE*). The staff service charge (*SSC*) is generally distributed evenly amongst staff of participating businesses (e.g., hospitality, administration and maintenance staff), and not only to the teams involved in MRW tourism (e.g., dive instructors, boat captains). The staff service charge (*SSC*) is similar to a mandatory gratuity and is paid out to staff on top of the base salary.

MRW employment revenue for 2021 was estimated as the combined value of staff salaries (*SSR*; Equation 3) using the number of local Maldivian *(M)* and foreign *(F)* employees *(E)* that work regularly on MRW focused trips including the following positions: snorkel/dive guides, boat crew/captain, administration staff and MPA management staff. The weekly staff wage (*SW*, US$) was conservatively assumed as US$164.3 for Maldivian *(M)* and US$201.2 for foreign *(F)* staff following Zimmerhackel et al. [52]. The annual staff salary was calculated based on the number of weeks per year (*NW*, weeks-1) dedicated to MRW activities. When the number of tour operator employees (*M* or *F*) were missing, a mean value based on the tour operator type was assigned.

$$SSR = SW_F E \times NW + SW_M E \times NW \tag{3}$$

The revenue generated from the management of the Baa Atoll Biosphere Reserve, specifically the Hanifaru MPA (*MPAR*), was used to assess the economic benefits of managing MRW tourism [85]. The revenue generated for management of the Hanifaru MPA (*MPAR*, US$) included resident (*EF_R*, US$) and tourist visitor (*EF_V*, US$) entry fees, videography permit fees (*VP*, US$), tour guide licence fees (*TGL*, US$) and professional partnerships fees (*PP*, US$) for streamlined site access (Equation 4).

$$MPAR = EF_R + EF_V + VP + TGL + PP \tag{4}$$

To estimate the revenue generated from tourist visitors at Hanifaru MPA ($REV_{HB}$, US$) by tour operators and management; the mean activity price of a snorkel in Baa ($AP_{BA}$, US$) and the number of tourist visitor entry fees ($N \times EF_V$, unitless) was used (Equation 5). This revenue excluded residents ($R$; Maldivian and work-permit holders) and their entry fees as these are either discounted or not charged by the resorts and activity centres.

$$REV_{HB} = AP_{BA} \times (N \times EF_V)$$

(5)

The economic indicators in this study reflect direct financial flows providing a measure of the industry's scale comparable to GDP. However, these equations do not represent the total economic value of MRW tourism, as they exclude consumer surplus, use and non-use values [86] and producer surplus (profits), and only represent the revenue generated [52]. The national contribution of both the tourism [49,50] and fisheries industries [87] were compared to the GDP of the Maldives. All values used in this study were converted to US$ for consistency. Although the market exchange rate from Maldivian Rufiyaa (MVR) to United States Dollar (US$) on 01/01/2022 was 0.064826 [88], we used the purchasing power parity which was 8.2107 in the Maldives in 2021 [89]. While the market exchange rate highlights the industries' role in the global market, the purchasing power parity-adjusted valuation underscores a higher domestic impact by considering the differences in the cost of living.

## Lifetime value of a manta ray

For both *M. alfredi* and *M. birostris*, the lifetime value ($LV$, US$) of an individual animal for both manta ray species over its lifetime in tourism (Equation 6) was calculated using: i) the estimated population size ($P$, unitless); ii) the estimated lifespan ($LS$, years); iii) the atoll-specific tour operator revenue ($TOR_{AS}$, US$) and tourist expenses ($TE_{AS}$, US$) for MRW in 2021, consistent with the criteria used in other studies [3,4] (Equation 6).

$$LV = \frac{TOR_{AS} + TE_{AS}}{P \times LS}$$

(6)

We adopted a non-discounted summation approach to estimate the cumulative tourism contribution attributed to an individual manta ray over its estimated lifespan. This method allocates total annual manta-related tourism revenue across the estimated population size and sums the resulting annual individual contribution over time. This approach was chosen to ensure direct comparability with previous elasmobranch tourism valuation studies [3,44]. We acknowledge that this method does not apply a discount rate to future revenues and does not incorporate the opportunity cost of natural capital. Consequently, the resulting estimates represent an undiscounted cumulative gross tourism contribution per individual, rather than a net present value or a full economic valuation.

The population of *M. alfredi* was assumed to be 3,500 individuals [55,56] and for *M. birostris* 1,000 individuals [55]. Both populations are a mixture of adult and juvenile individuals (reaching maturity at 8–17 years of age [65]), with mostly adults visiting diving sites and a mixture of both at snorkelling sites [55,57]. The estimated lifespan for *M. alfredi* is ~40 years [90], and because the lifespan for *M. birostris* is unknown, we assumed it to be equal to that of *M. alfredi*. While both *M. alfredi* and *M. birostris* frequent Maldivian waters, their distributions differ significantly; notable, Fuvahmulah is the only region where *M. birostris* is reliably encountered [57]. To calculate the lifetime value ($LV$) for each species, it was necessary to disaggregate the combined revenue into atoll-specific visitation metrics ($TOR_{AS} + TE_{AS}$). We applied distinct attribution logic for fixed-based operators (resorts, community islands) and mobile bases (liveaboards). Because liveaboards traverse multiple atolls during a single trip, while fixed-base operators generally operate within a single region, revenue were weighted on reported spatial effort. Consequently, the revenue attributed to *M. birostris* was restricted to tour operators in Fuvahmulah only, while revenue for *M. alfredi* was derived from seasonal visitation across all other regions. Two liveaboard tour operators who completed the survey ($n = 8$) highlighted that they visited Fuvahmulah Atoll in the Northeast

Monsoon. Also in the same season, Manta Trust's *Maldives Manta Conservation Programme* long-term monitoring data-base revealed regular presence of liveaboards in routine surveys [55,91]. We estimated that 4% of the revenue generated ($TOR_{AS}$ + $TE_{AS}$) from liveaboards was from time spent in Fuvahmulah and the other 96% of revenue was from the remaining regions, based on the responses on visitation frequency from eight tour operators.

### Intrinsic value of manta rays

The importance of manta rays for business operations, local communities, and education outreach were valuated by quantifying the frequency of the words used to answer three open-ended questions in the tour operator surveys, including: 1. Why/why-not do you consider manta rays to be important to your business; 2. Why/why-not do you consider manta rays to be important to local communities in the Maldives; and 3. Do you think that your manta ray trips helped to educate guests about manta rays and conservation, and how? These data were processed through systemising descriptive responses to identify patterns and themes behind textual data. Word count analysis was used to show the most frequently used words to describe the importance of MRW tourism by survey respondents. Words count analysis and the removal of stop words was done using the 'tidytext' R package [92]. Some of the words used in the questions were also removed pre-analysis, e.g., "manta", "mantas", "manta ray", "Maldives". Additionally, eight visitor surveys conducted by the Ministry of Tourism between 2011–2021 (visitor participants $n = 11,625$) were used to assess the primary motivations of tourists, visitor satisfaction and diving/snorkelling participation [93].

## Results

### Scale of manta ray watching tourism

A total of 80% ($n = 282$) of the 354 tourism bases in the Maldives offered MRW diving and/or snorkelling in 2021 (Table 1). Within these tourism bases, 69% ($n = 374$) of the 538 registered tour operators participated in MRW tourism. The MRW industry is widely distributed across the Maldives, being present in all but one of the 20 administrative regions used in this study. However, land-based experiences were concentrated primarily in Kaafu, Alif Dhaal and Baa, which had 76, 46, and 43 tour operators, respectively. Whereas liveaboard experiences generally visited multiple regions, with Alif (both Alif Alif and Alif Dhaal), Kaafu and Meemu being the most visited during the NE Monsoon, and Baa, Kaafu and Alif in the SW Monsoon.

**Table 1. Registered, verified, and operational tour operators in the Republic of Maldives (established by the Ministry of Tourism and verified by this study; *n* = 538) and the extent of manta ray watching (MRW) tourism.**

| Tour operator type | Base type [*n*] | Tour operators [*n*] | Tour operators offering MRW [*n* (%)] | Tour operators (separated into activity centre type) offering MRW [*n* (%)] | | Tour operators that completed the survey [*n* (%)] | Survey participants – offering MRW [*n* (YES)] |
|---|---|---|---|---|---|---|---|
| | | | | Dive Centre | Excursions Centre | | |
| 5-star luxury resorts | Private resort island (n = 48) | 96 | 62 (65%) | 35 (73%) | 27 (56%) | 21 (22%) | 20 |
| Regular resorts | Private resort island (n = 118) | 236 | 139 (59%) | 80 (68%) | 59 (50%) | 43 (18%) | 42 |
| Local centres | Inhabited community island (n = 40) | 107 | 88 (82%) | 66 (80%) | 16 (94%) | 29 (27%) | 27 |
| Liveaboards | Vessel (n = 99) | 99 | 85 (86%) | 85 (86%) | | 13 (13%) | 12 |
| **Total** | | 354 | 538 | 374 (70%) | | | 106 (20%) | 101 |

Tour operators reported 92 MRW sites for diving/snorkelling with manta rays. Within these sites, 65 were regularly visited for diving and 56 for snorkelling (some sites were used for both activities). They identified Hanifaru MPA in Baa ($n = 18$) and Lankan Beyru in Kaafu ($n = 17$) as the two primary MRW sites (Table 2). Hanifaru MPA was the most important MRW site in terms of number of tourists counted at the time of visitation during a trip ($n = 52.6 \pm 6.1$ Standard Error (SE)), number of boats counted during a trip ($n = 8.8 \pm 1.2$ SE), and the number of manta rays sighted during a trip (snorkelling trips only) ($n = 32.3 \pm 4.0$ SE). Most of the MRW sites were primarily frequented by *M. alfredi*.

## Economic benefits from manta ray watching tourism

Tour operator revenue (*TOR*) for manta-ray-specific snorkelling, diving, and diving on a liveaboard in the Maldives in 2021 was estimated to be US$39,027,696 (Table 3) from 475,061 guests participating in MRW experiences (diving $n = 350,689$, snorkelling $n = 124,372$). Most of the TOR revenue is generated from MRW experiences in three regions, including Kaafu, Alif Dhaal and Baa, which generate approximately 12% (US$4.8 million), 15% (US$5.8 million), and 18% (US$6.9 million) of the total tour operator revenue (*TOR*) respectively (Fig 2). Five-star resorts contributed the least to the tour operator revenue when compared with other tour operator types (Fig 3e). Despite having the highest activity prices (*AP*), 5-star resorts had the lowest number of guests per trip (*NG*), trips per week (*NT*), and season length (*NW*; Fig 3a–d). The mean activity price (*AP*) across operators was US$86.6 ± 2.3 SE for diving and US$121.8 ± 6.6 SE for snorkelling. The perceived value to businesses of an opportunistic encounter with a manta ray was high, as demonstrated by 81% of tour operators that featured manta rays in their operation's marketing ($n = 448$; i.e., logo, photos on website, marketing mentions famous manta ray dive sites).

Table 2. Most visited manta ray watching (MRW) dive and snorkel sites as ranked by tour operators listed in their top five, and the mean (± SE) number of tourists, tourist vessels and manta rays per trip. The active MRW season is categorised as either the northeast (NE) or southwest (SW) monsoon.

| Region | Manta ray watching site name | Latitude (°) | Longitude (°) | Species primarily encountered | Number of tour operators that listed the site [$n$] | Monsoon active | Activity | Mean no. tourists per trip [$n \pm SE$] | Mean no. manta rays per trip [$n \pm SE$] | Mean no. boats per trip [$n \pm SE$] |
|---|---|---|---|---|---|---|---|---|---|---|
| Baa | Hanifaru MPA | 5.17 | 73.15 | *M. alfredi* | 18 | SW | Snorkelling | 52.6 (± 6.1) | 32.3 (± 4.0) | 8.8 (± 1.2) |
| Kaafu | Lankan Beyru | 4.28 | 73.56 | *M. alfredi* | 17 | SW | Both | 26.0 (± 4.5) | 4.1 (± 0.4) | 5.1 (± 0.8) |
| Raa | Sola Corner | 5.49 | 72.83 | *M. alfredi* | 12 | SW | Both | 15.3 (± 4.5) | 4.0 (± 0.3) | 3.1 (± 0.3) |
| Alif Dhaal | Rangali Madivaru | 3.59 | 72.72 | *M. alfredi* | 12 | NE | Both | 32.4 (± 7.3) | 4.8 (± 0.4) | 6.8 (± 1.3) |
| Baa | Dhonfanu Faru | 5.18 | 73.12 | *M. alfredi* | 11 | SW | Both | 22.5 (± 7.5) | 11.4 (± 4.9) | 4.4 (± 1.1) |
| Raa | Kottefaru Thila | 5.51 | 73.03 | *M. alfredi* | 11 | NE | Both | 10.0 (± 2.1) | 6.3 (± 1.0) | 2.8 (± 0.3) |
| Alif Dhaal | Moofushi Bojamhadi | 3.88 | 72.71 | *M. alfredi* | 11 | NE | Both | 35.8 (± 5.9) | 4.6 (± 0.7) | 3.4 (± 0.4) |
| Baa | Dharavandhoo Corner | 5.16 | 73.14 | *M. alfredi* | 10 | SW | Both | 13.3 (± 0.9) | 3.0 (± 0.0) | 3.5 (± 0.2) |
| Fuvahmulah | Farikede Faru | −0.32 | 73.45 | *M. birostris* | 10 | NE | Diving | 15.7 (± 6.7) | 3.8 (± 1.4) | 2.5 (± 0.6) |
| Kaafu | Guraidhoo Kandu | 3.89 | 73.47 | *M. alfredi* | 10 | SW | Diving | 13.5 (± 2.9) | 3.3 (± 0.4) | 2.3 (± 0.5) |

**Table 3. Economic benefits of manta ray watching (MRW) tourism in the Republic of Maldives in 2021 collected via internet research and tour operator surveys. Tourist expenses (*TE*) are additional costs and are not included in the tour operator revenue (*TOR*). The 374 tour operators that offer MRW tourism are categorised based on the activity type they offer (diving-specific operators n = 267 and snorkelling-specific operators n = 107). Tour operators sold 475,061 tickets to take guests on a MRW trip (diving-specific trips n = 350,689 and snorkelling-specific trips n = 124,372) (All values in US$).**

| | Revenue source | Diving-specific revenue (*D*) (US$) | Snorkelling-specific revenue (*S*) (US$) | Total (US$) |
|---|---|---|---|---|
| $TOR_{DI/SI/LAB}$ | Tour operator revenue | $25,929,181 | $13,098,515* | $39,027,696 |
| *TE* | Tourist expenses | $97,418,749 | $90,851,933 | $188,270,681 |
| *TX* | Tax | $18,502,189 | $15,592,567 | $34,094,757 |
| *SSC* | Staff service charge | $12,334,793 | $10,395,045 | $22,729,838 |
| *SSR* | Employee staff salaries | $21,636,162 | $5,352,606 | $26,988,768 |
| | **Sub total** | $175,821,074 | $135,290,665 | $311,111,739 |
| *MPAR* | Management | | $252,891 | $252,891 |
| $DEB_{MRW}$ | **DIRECT economic benefit** | $175,821,074 | $135,543,556 | $311,364,630 |

*The Hanifaru MPA ticket revenue was deducted from tour operator revenue as it is accounted for by the MPA management (*MPAR*).

Benefits from MRW tourism also reach other local businesses. Tourist expenses (*TE*) from MRW dive/snorkel guests were an estimated US$188,270,681 for local businesses, including accommodation, food and beverages (Table 3). The combined revenue (*TOR* + *TE*) of MRW was estimated at US$227,298,377 and contributed an estimated 2.6% to the annual 2021 GDP purchasing power parity in the Maldives (Table 4). Tourism constitutes a significant portion of the Maldivian economy, contributing over 25% to its GDP in 2021 [49].

Socio-economic benefits of MRW tourism flow into the community in the form of staff salary revenue (*SSR*). The tour operator surveys revealed that operators employ 3,876 staff (Maldivian n = 2,602, Foreign n = 1,262) who regularly work on MRW trips (Table 5). In addition, 12 Maldivians were employed to manage the Baa Atoll Biosphere Reserve [85]. The mean annual salary for Maldivian staff (US$8,544) was 81.7% that of foreign staff (US$10,464). Calculated only for the number of weeks (*NW*) of the MRW season, the total staff salary revenue (*SSR*) for tour operator staff was estimated at US$26,886,240 (Maldivian = US$14,675,591, Foreign = US$12,210,649) and for the Biosphere Reserve staff (Maldivian = US$102,529). Local Maldivian staff make up 67% of the workforce employed to work in MRW tourism and management and receive 55% of the distributed staff salary revenue (*SSR*). The flow-on benefits of the MRW industry additionally extend to the government through tax (*TX*; US$34,094,757) and then again to the workforce (not only those employed in roles to directly work in MRW tourism) in the form of staff service charge (*SSC*; US$22,729,838; Table 3).

In terms of management, the single monitored and ticketed MPA in the Maldives, Hanifaru, in the Baa Atoll Biosphere Reserve (*MPAR*), generated US$252,891 in revenue for management bodies in 2021 (Table 6). The *MPAR* includes the visitor ($EF_V$) and resident entry fees ($EF_R$), videography permit fees (*VP*), tour guide licence fees (*TGL*) and professional partnerships (*PP*). This MPA was the top visited MRW site in the Maldives, as indicated by tour operator surveys (Table 2). On average, tour operators charged US$137.3 ± 14.3 SE per snorkel trip (*AP*) in Baa. The 11,258 visitors to Hanifaru MPA generated an estimated US$1,319,171 in revenue for tour operators and management ($REV_{HB}$). The overall direct economic benefits ($DEB_{MRW}$) of MRW tourism in the Maldives in 2021 was estimated at US$311,364,630 (Table 3).

## Lifetime value of a manta ray

The lifetime values (*LV*) are undiscounted cumulative gross tourism contributions per individual over an assumed 40-year biological lifespan. These were calculated using the atoll-specific generated revenue; for *M. birostris* using revenue from Fuvahmulah ($TOR_{AS}$ = US$705,036 and $TE_{AS}$ = US$1,575,327), and for *M. alfredi* using the revenue from the other

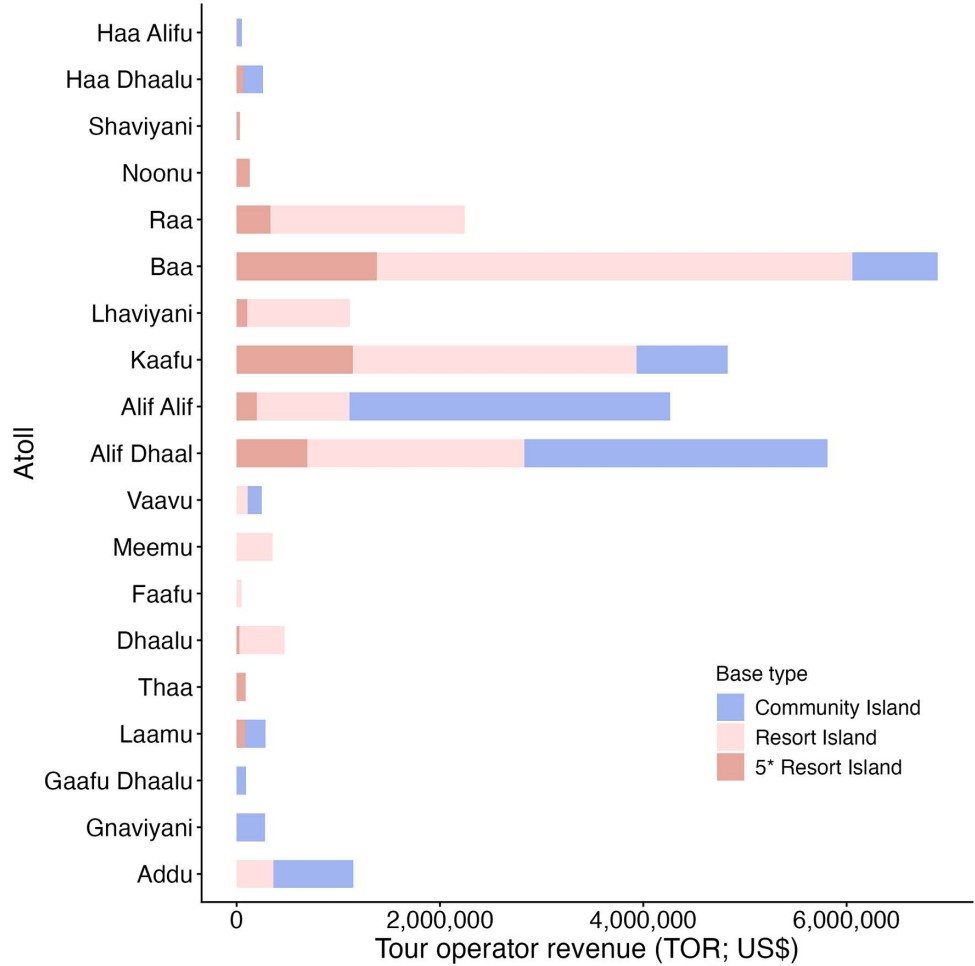

**Fig 2. Estimated tour operator revenue (*TOR*) and distribution of manta ray watching tourism in the Republic of Maldives across tourism bases (all values in US$).** The land-based tour operator revenue (TOR) excludes liveaboards as they generally visit multiple regions during a single trip. Each bar represents a single administrative region (*n* = 19). Regions are ordered from north (Haa Alif) to south (Addu). Malé City is grouped with Kaafu. There is no reported manta ray watching tourism in Gaafu Alifu, thus it did not generate revenue, and it is not displayed.

regions ($TOR_{AS}$ = US$38,322,660 and $TE_{AS}$ = US$186,695,354). An individual *M. alfredi* was estimated to be worth US$2,571,634, compared to a value of US$91,215 for *M. birostris* during their lifetime (Fig 4).

### Intrinsic value

Tour operator survey respondents provided insight into the intrinsic value of manta rays to their businesses, to in-country visitors, and to local communities (Fig 5). All tour operators ranked manta rays in the top five sea life that visitors most wanted to see (*n* = 100) with 28% ranking them as the top attraction, 47% ranking them in their top two, 19% in their top three and 6% in their top four. Operators perceived manta rays to be important to their business (yes = 97%, *n* = 101) and to the wider local community (yes = 99%, *n* = 101; Fig 5). The word count analysis showed the most frequently used words by survey responders to describe why they believed manta rays to be important to their operation (n = 103) were: "guests" (*n* = 28), "people" (*n* = 18), "season" (*n* = 10), "divers" (*n*=8), resort" (*n*=8) and "trips" (*n*=8). It was also reported that manta rays in the Maldives were a "bucket list" item for many guests (*n*=9). Compared to when survey respondents were asked

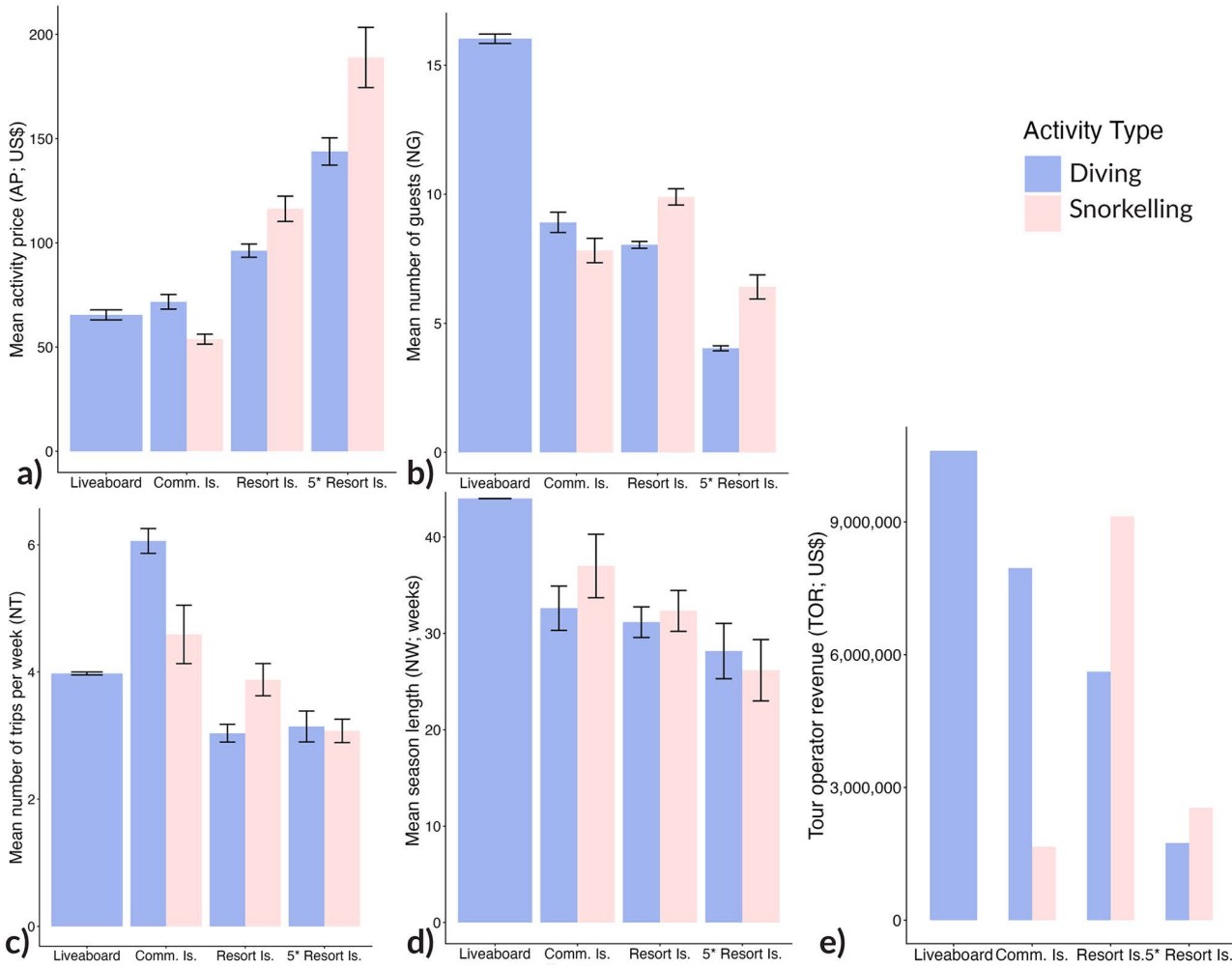

**Fig 3. Tour operator revenue (*TOR*) generated through manta ray snorkelling and diving activities categorised by tour operator type in the Republic of Maldives in 2021.** The tour operator revenue (TOR) was calculated as a product of a) activity price per guests (*AP*), b) number of guests per trip (*NG*), c) number of trips per week (*NT*) and d) season length (in weeks) per year (*NW*), displayed here as the Mean (± SE) value (All values in US$). Comm. Is., community Island; Resort Is., resort island; 5*Resort Is., 5-star resort island.

to describe the importance of manta rays to the community (*n* = 102), the most frequently used words were: "local" (*n* = 25), "tourism" (*n* = 24), "income" (*n* = 16), "communities" (*n* = 15) and "tourists" (*n* = 12). One survey respondent also reported that "Mantas have a certain cultural significance to local communities in the Maldives as well as attracting tourists to the area along with the associated economic benefits".

The value of manta rays was additionally demonstrated with 61% of survey respondents (*n* = 56) reportedly offering an educational presentation and/or briefing to trip guests through their operation (Fig 5). These presentations generally covered aspects of biology, threats to and conservation of manta rays, and how to swim responsibly with these animals. Without taking into consideration tour operators that did not complete the survey, it is estimated that at least 17,655 MRW guests received a presentation or briefing as part of their MRW trip experience in 2021. Most tour operators believed that taking guests to see manta rays helped to educate and conserve these animals (95%, *n* = 95). The most frequently used words by survey respondents to describe why they believed that manta ray trips helped to educate guests about manta

**Table 4.** The Gross Domestic Product (GDP) in the Republic of Maldives and the share contribution of national industries including manta ray watching (MRW) tourism for 2008–2010, and 2019–2021. The GDP and Tourism Contribution values were converted from Maldivian Rufiyaa (MVR) to United States Dollar (US$) for consistency using the purchasing power parity which was 8.2107 in the Maldives in 2021 [89].

| Year | Number of tourists (n) | GDP Purchasing power parity (US$)[a] | Tourism Contribution (US$) [a] | Share of Tourism Contribution (%) [a] | Fisheries Contribution (US$) [b] | Share of Fisheries Contribution (%) [b] | MRW tour operator revenue Contribution (US$) | Share of MRW tour operator revenue Contribution (%) | MRW combined revenue Contribution (US$) | Share of Combined economic revenue Contribution (%) |
|---|---|---|---|---|---|---|---|---|---|---|
| 2008 | 638,012 | $2,256,351,546 | $593,135,703 | 26.3 | | | $8,100,000 [c] | 0.4 | $15,471,000 [d] | 0.7 |
| 2009 | 655,852 | $2,149,416,608 | $561,225,733 | 26.1 | | | | | | |
| 2010 | 791,917 | $2,553,528,366 | $649,818,527 | 25.4 | | | | | | |
| 2019 | 1,702,887 | $9,407,107,885 | $2,453,048,498 | 26.1 | $285,801,281 | 3 | | | | |
| 2020 | 555,494 | $6,256,424,622 | $907,972,621 | 14.5 | $315,336,273 | 5 | | | | |
| 2021 | 1,321,937 | $8,868,170,414 | $2,288,017,928 | 25.8 | $307,748,520 | 3.5 | $39,027,696 | 0.4 | $227,298,377 | 2.6 |

Source

[a]Ministry of Tourism [49]

[b]Bureau of Statistics [87]: The percentage calculations for Tourism and Fisheries contributions were taken directly from the Bureau of Statistics reports and not calculated.

[c]Anderson et al. [44]

[d]O'Malley et al. [3]

**Table 5.** Staff employed in operations that actively participate in manta ray watching tourism. Calculations only account for respective manta ray watching seasons.

| Code | Salaries | Foreign Staff [n (F)] | Maldivian Staff [n (M)] | Total Staff (n) |
|---|---|---|---|---|
| E | Number of employees working with MRW diving | 1,024 | 1,893 | 2,781 |
| E | Number of employees working with MRW snorkelling | 238 | 709 | 1,083 |
| E | Number of employees working in Management | 0 | 12 | 12 |
| | **Total employees** | 1,262 | 2,614 | 3,876 |
| SW | Weekly salaries (US$) | **$201.23*** | **$164.31*** | |
| *SSR* | **Total staff salary revenue (US$)** | $12,210,649 | $14,778,119 | $26,988,768 |

*Weekly salaries were obtained from Zimmerhackel et al. [52].

**Table 6.** Revenue generated for management bodies from Hanifaru Marine Protected Area (*MPAR*).

| Code | Category | Number (N) | Price per unit (US$) | Revenue generated (US$) |
|---|---|---|---|---|
| $EF_R$ | Resident & work-permit visitor entry-fee | 690 | $2.10 | $1,431 |
| $EF_V$ | Tourist visitor entry fee | 11,258 | $20.10 | $226,245 |
| TGL | Tour-guide licence fee | 75 | $19.45 | $1,459 |
| PP | Professional partnerships | 11 | $993.75 | $10,931 |
| VP | Videography permit | 11 | $1,166 | $12,826 |
| *MPAR* | **Total** | | | $252,891 |

Amount (N) and price per unit sourced from Baa Atoll Biosphere Reserve [85].

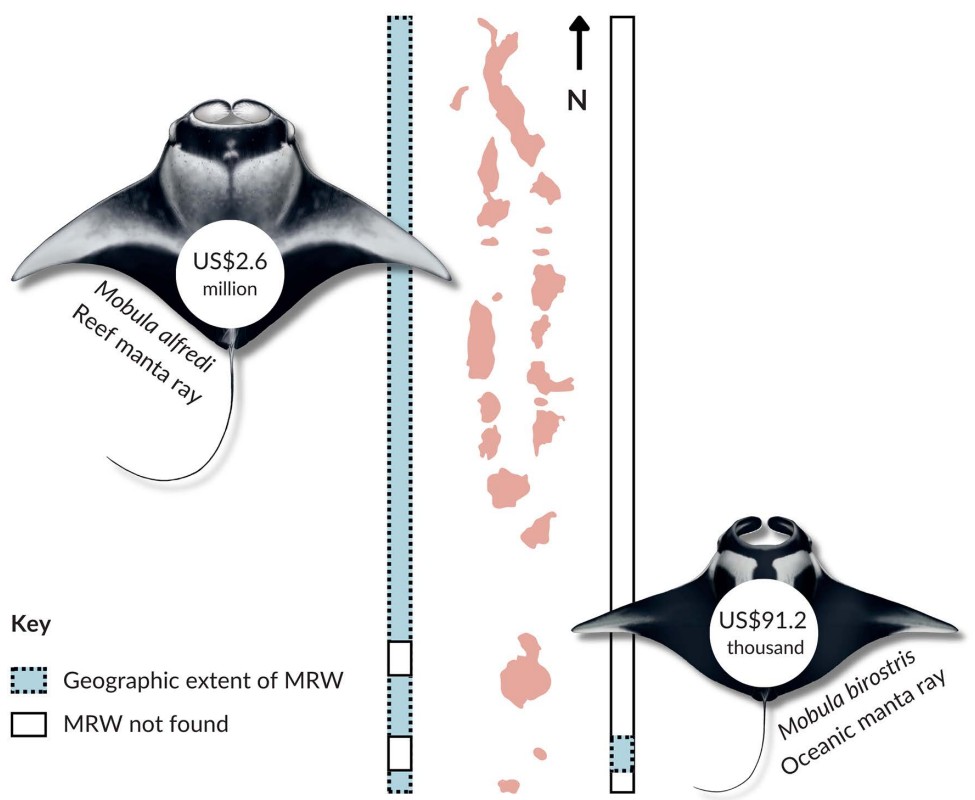

**Key**

▨ Geographic extent of MRW

▢ MRW not found

**Fig 4. The lifetime value of individual manta rays (both species) as part of tourism calculated using the combined economic revenue from manta ray watching in the Republic of Maldives in 2021.** Tourism that focused on Mobula alfredi (reef manta rays) was identified in 18 administrative regions, compared to only one region for *M. birostris* (oceanic manta rays; Gnaviyani/Fuvahmulah). The geographical extent of manta ray tourism for each species is shown using coloured dotted rectangles.

rays and conservation ($n = 100$) were: "guests" ($n = 26$), "behaviour" ($n = 19$), "briefing" ($n = 15$), "information" ($n = 14$), and "conservation" ($n = 13$).

## Discussion

### Economic benefits of manta ray watching tourism

We demonstrate that marine wildlife tourism provides immense value to local and national economies by generating revenue, supporting cultural heritage, and creating employment opportunities. Using the Maldives as a case study, this study provides an updated national valuation of the economic benefits of MRW tourism. We found that the 475,000 dive and snorkel trips taken by MRW guests in 2021 generated an estimated US$227.3 million, comprising US$39 million in revenue for tour operators and US$188.3 million in revenue of secondary tourist expenditures. These estimates represent a substantial increase from previous assessments in 1997 (US$7.8 million) and 2008 (US$8.1 million) [43,44]. When the industry was last valued at a national scale in 2011 (using 2008 data), it was estimated to generate US$15.5 million in combined revenue, including US$7.4 million in tourist expenditures derived from the 157,000 MRW trips [3].

While these estimates suggest significant expansion, direct comparisons are complicated by evolving methodologies. The 2011 study [3] utilised a site-density approach for MRW sites, whereas our study employed a high-resolution approach combining tour operator surveys and online data mining. Both estimates are inherently conservative, accounting

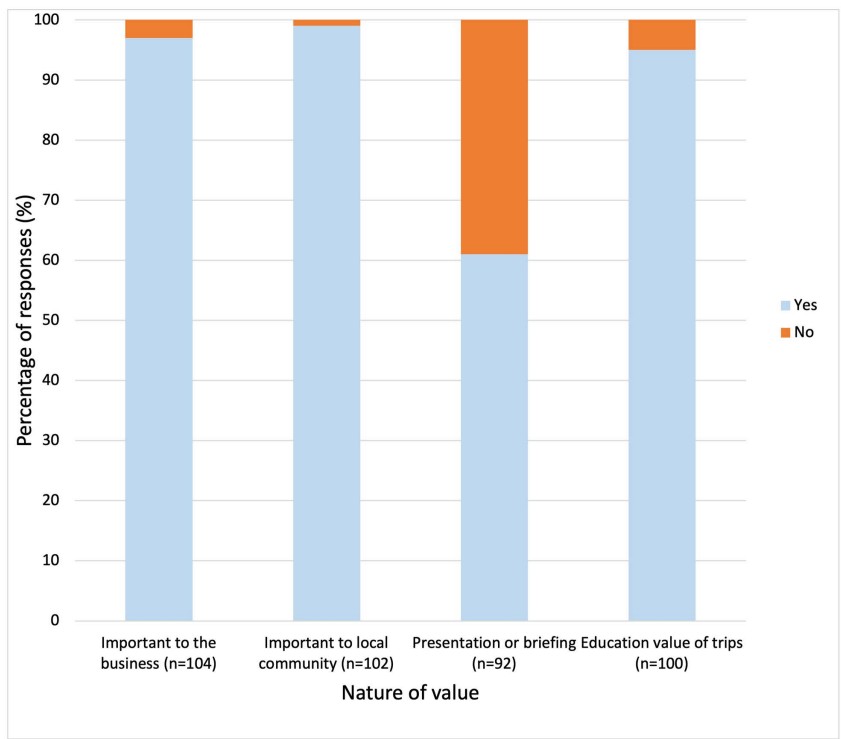

**Fig 5. The perceived socio-economic importance of manta ray watching tourism to tour operators (proportion) in the Republic of Maldives.**
These responses were via survey questions about the nature of the value, e.g., to their business, the community, and as a tool for education and conservation for guests.

only for quantifiable metrics; however, the 2011 estimate was limited by its focus on tourism density at known MRW sites, and the benefit transfer ratio used to calculate secondary tourist expenses did not account for pricing tiers between operator types (e.g., expenses at a luxury resorts vs at a community island) [3]. Further, our 2021 estimate may be influenced by broader market inflation and a more comprehensive capture of secondary supply chains that were previously unmonitored.

Despite these methodological differences, the 2021 value reflects a remarkable >380% growth in tour operator revenue over the past decade. The growth underscores the increasing economic significance of MRW tourism and mirrors the trajectory of the broader Maldivian tourism sector, which reported a 286% increase in GDP contribution over the same period (2008 = US$593.1 million and 2021 = US$2.3 billion) [49]. Consequently, while the different methods utilised over the last two decades prevent precise direct comparisons of absolute values, the overarching trend clearly establishes MRW tourism as a cornerstone of the national economy.

Alongside the growth in revenue from tourism, the annual international visitor numbers to the Maldives have doubled (2008 = 638,012 and 2021 = 1,321,937) [49]. During routine departure surveys from 2009 – 2022 ($n = 11,625$), over half (55%) of the international visitors cited the "underwater beauty" of the marine environment as the primary reason for their in-country visit [93]. Many international visitors were also specifically attracted by diving and snorkelling (26%), and particularly by megafauna, including manta rays (22%) [93]. Reef sharks and whale sharks are also economically important elasmobranch megafauna in the Maldives, with both industries reporting revenue growth over recent years [39,52]. Although mindful of methodological differences, reef shark diving contributed US$14.4 million in tour operator revenue in 2016, an 83% increase from US$7.9 million in 1992 [43,52]. While whale shark watching contributed US$9.4 million in

2013, rising from US$7.6 million in 2012 [39], the industry therefore growing 24% in one year (no difference in methods). Although reef and whale shark tourism in the Maldives show industry specific growth, they do not demonstrate growth of a similar magnitude to MRW tourism.

Tour operators ranked manta rays in the top five sea life that diving and snorkelling tourists wanted to see, with 28% ranking them as the top attraction. Manta rays also generate value through marketing and are used as a selling point for businesses to attract potential customers. Despite some operators not directly selling manta-ray-focused trips, the perceived value of even an opportunistic encounter holds substantial business appeal, highlighting the profitability of promoting MRW tourism. Consistent with the increased number of international visitors, the Maldives reported a 272% increase in registered bases (including guesthouses) from 250 in 2008, to 929 in 2021 [49,50]. Moreover, during the same timeframe, the mean price of a MRW diving trip nearly doubled, from US$45–70 to US$87, and snorkelling trips saw a six-fold increase from US$20 to US$121.8 [44]. The large increase in the price for a MRW snorkelling trip could be attributed to the rise in popularity of this kind of activity, particularly at resort bases which are generally more expensive [82].

MRW tourism contributed substantially to the Maldives' economy, representing 2.6% of its GDP in 2021, similar to that of the fishing and fish processing industry, which contributed 3.5% to the GDP in the same year [87]. The Maldives is a tuna fishing nation [94] and has an especially high dependence on the sustainable availability of marine resources for both food security and economic prosperity [33]. As a country that does not harvest its manta rays [36], but is reliant on both fisheries and tourism industries, the Maldives provides an example of how these two industries can co-exist when managed sustainably. This contrasts with regions such as Mozambique, where manta ray sightings in one tourism hub decreased by 92.5–99% over two decades, largely due to fisheries mortality [95]. While specialized elasmobranch fisheries often operate offshore and target specific size classes—meaning they do not always overlap spatially with tourism sites—the resulting population-level declines can eventually impact sighting rates at dive locations [3].

Similar tensions between tourism and unsustainable harvesting exist in other biodiversity hotspots like Indonesia and the Philippines [3]. These regions face the complex challenge of balancing lucrative elasmobranch tourism with active local fisheries. However, management scales and structures differ significantly across these locations. For instance, the Raja Ampat region in Indonesia encompasses a 40,000 km² Regency where a network of five MPAs is managed by a localised management unit [96]. This decentralised approach allows for regional oversight of its 51 known MRW sites, contrasting with the more centralised management framework of the Maldives' MPA network. Understanding these diverse management models is critical for replicating the economic success of the Maldives in other regions facing similar threats.

The contribution of MRW to the Maldives economy exceeds estimates from older studies in other nations that used similar survey design and analysis [3,8,44]. Nevertheless, the comparison between Mozambique in 2014 (US$34 million; [8]), the global industry in 2012 (US$140.7 million; [3]), and the Maldives in 2021 (US$227.3 million) demonstrates the huge size of the Maldives industry. In a global review of MRW tourism by O'Malley et al. [3], the Maldives had the highest number of both MRW sites and guests. Given the current estimate for 2021, it is likely that the Maldives remains one of the highest, if not the highest, MRW revenue generating nations world-wide.

Both the Maldivian economy and the livelihood of its people are largely dependent on marine resources [97]. This study provides evidence for the flow-on socio-economic impacts of manta ray tourism in the Maldives. Almost 4,000 staff were employed by tour operators to work directly on MRW diving and snorkelling tourism activities, generating US$27 million per year in staff salary revenue. Despite the 2,602 Maldivian staff representing 67% of the MRW tourism workforce, they only received 55% of the distributed staff salary within the industry. A similar discrepancy was observed in reef shark-diving tourism in 2016, where Maldivians represented 55% of the workforce but received 50% of the distributed staff salary [52]. The wage gap likely stems from the disproportionate employment of Maldivian nationals in lower-paid operational roles, such as divemaster, equipment maintenance, or boat operations, while expatriates more frequently occupy senior management and instructional positions, a pattern observed in many Global South tourism industries [98,99].

While some operators in the Maldives are already implementing initiatives to further address these disparities, encouraging the widespread adoption of a structured approach toward localisation remains vital. Beyond fostering individual career progression, these initiatives offer broader benefits, including enhanced local economic development, long-term economic sustainability, and reduced recruitment costs [98,100]. Formalising and rewarding the high-value traditional ecological knowledge possessed by Maldivian guides with competitive, senior-level wages is essential for ensuring that marine tourism offers a lucrative and prestigious career path. Maldivians often possess knowledge regarding manta ray aggregations and oceanographic patterns that is indispensable for high-quality guest experiences. If the tourism sector fails to provide a career trajectory that competes with other high-paying industries, this human capital may be lost to sectors that do not contribute to the blue economy or conservation. By incentivising the promotion of locals into specialist roles, the Maldives can ensure that the economic benefits of its natural wealth are not only generated but also equitably retained within its communities [98,100].

Socio-economic benefits also flow-on to reach the government via tax and MPA management, and to staff employed in the hospitality and tourism industries through staff service charge. These diverse revenue streams underscore the industry's substantial local importance. In 2021, the economic and socio-economic benefits of MRW tourism to businesses, government and the community in the Maldives exceeded US$311 million. While detailed economic valuations of wildlife tourism that include many facets of direct contributions are rare in wildlife tourism studies, a comparable calculation for shark-diving tourism in the Maldives using similar methods (though excluding staff service charge), estimated overall direct economic benefits at US$77.1 million in 2016 [52]. The substantial difference in revenue highlights the particularly high economic contribution of MRW within the Maldives' tourism sector. The portion of the US$311 million in revenue generated by MRW tourism that stays in-country and is distributed for environmental and biodiversity conservation remains unknown. However, there is disparity between the amount of revenue generated through nature-based tourism in the Maldives and the portion of the revenue that is used by government agencies for conservation [97]. More than half of the funding that supports in-country conservation comes from unstable international sources (US$16.3 million in 2013) [97].

## Scale and potential negative impacts of tourism

While concentrated in certain regions (i.e., Alif Alif/Dhaal, Baa, Lhaviyani, Kaafu) [52], diving tourism, particularly MRW, has undergone rapid expansion across the Maldives [49,77,101] raising concerns about the sustainability and management of the MRW tourism industry. In 2008, the Maldives had 101 MRW sites (dive sites $n = 91$, snorkel sites $n = 10$), representing nearly half (48%; 91 of 190 sites) of all known global MRW sites at the time [3]. By 2017, the amount of recognised sites surged to 273 [58]. Our 2021 findings further demonstrate continued growth, with MRW offered by 80% of tourism bases and present in 95% of administrative regions. Tour operators in this study identified a total of 92 sites they considered top five for diving ($n = 65$) and snorkelling ($n = 56$) with manta rays. Some sites in the regions surrounding the central Malé City received a considerably high portion of the tourism pressure from diving, snorkelling and vessels (i.e., Hanifaru in Baa, Lankan Beyru in Kaafu, Sola Corner in Raa, and Rangali Madivaru in Alif Dhaal). As the top visited site in 2021, the > 11,000 annual visitors to Hanifaru generated US$1,514,400 in revenue by tour operators and the Baa Atoll Biosphere Reserve Office. Although Hanifaru does fall within an MPA and is actively monitored and mitigated by regulatory components [36], this does not exclude this site from the potential adverse impacts of wildlife tourism activities on the focal species and their associated habitats [69,76,102].

When tourism is unmanaged or is beyond the carrying capacity of an area, the industry is unsustainable and requires further regulation and enforcement. In the Maldives, sublethal injuries [70] and tourist behaviour [69] negatively impact manta rays and lead to a possible reduction in reproductive fitness [56]. These impacts can become more pronounced during the climate crisis due to a reduced availability of the manta ray's planktonic prey which is required to sustain reproductive health and overall fitness [62]. Other animal groups such as cetaceans have shown to react to tourist vessels with active avoidance strategies [103,104]. Such repeated disturbances even led to displacement from a preferred habitat and

reduced fitness at the population level for bottlenose dolphins (*Tursiops sp.*) [103]. The projected reduction in reproductive fitness of manta rays with exposure to threats highlight the importance of safeguarding known foraging areas and other critical habitats. Despite the expansive nature of MRW activities in the Maldives, the visitation density at these numerous sites remains unquantified. Investigating the crucial metric of visitation density in the future is essential for robustly assessing and managing the potential negative impacts of tourism.

## Management and sanctuaries

There has never been a targeted commercial fishery for manta rays in the Maldives; however, these populations are still vulnerable to bycatch, illegal fishing, habitat degradation, pollution, and unregulated tourism [36,69,70]. The Maldivian government's recognition of the economic value of iconic animals such as manta rays, whale sharks, and reef sharks, was instrumental in developing legal frameworks and policies such as the Hanifaru MPA, and the protection of shark and ray species [35,36]. Importantly, successful conservation relies on community support; a lack of buy-in and top-down conservation approaches can lead to non-compliance or policy reversal [105]. Recent attempts to reestablish longline fisheries and legalise shark fishing in the Maldives in 2024 threatened the sanctuary status, but were successfully blocked through community pushback and an online poll [106,107]. Yet in 2025, the government reopened the gulper shark (*Centrophorus* spp.) fishery, an action met with complex attitudes and strong community opposition [108]. These events underscore the need for robust conservation strategies that align national conservation goals with international commitments to the United Nations Convention on Biological Diversity (CBD) and the Convention on the Conservation of Migratory Species of Wild Animals (CMS).

Nationwide there are 42 MPAs in the Maldives covering 116.3 km$^2$ (0.5% of the 21,596 km$^2$ of Economic Exclusive Zone), yet for many years Hanifaru remained the only MPA with active, site-specific monitoring and enforcement [36,58,69]. The high density and predictability of manta rays at Hanifaru MPA (mean of 32 individuals per trip) is driven by the geomorphological and oceanographic conditions, eddies that trap high concentrations of their zooplankton prey, making it a globally unique aggregation [109]. Although such densities of manta rays infrequently occur elsewhere in the Maldives, the Hanifaru MPA model demonstrates how rigorous regulation can successfully mitigate the impacts of high visitor pressure [69]. Since its designation in 2009, regulations have limited congestion by capping the number of visitors at 80 snorkellers and five boats at any one time, prohibiting SCUBA diving, enforcing a code-of-conduct and requiring licensed guides [69,110]. In 2012, the entirety of Baa Atoll was formally designated as the Baa Atoll Biosphere Reserve, with Hanifaru MPA as a core protected zone. Revenue generated through the Biosphere Reserve supports the Baa Atoll Conservation Fund, which is supposed to be utilised for community education and research [110].

In 2025, drawing from the success of Hanifaru MPA, long-term monitoring and stakeholder consultation led to the introduction of active monitoring and enforcement of the South Ari Marine Protected Area (SAMPA), a critical aggregation area for whale sharks [111–113]. By implementing regulations and revenue-generating mechanisms modelled after Hanifaru MPA at other key megafauna aggregation sites, the government could prevent the degradation of high-traffic habitats while maintaining high-quality tourism experiences. A significant step towards this national scaling occurred in 2023 with the development of the Protected Species Regulation (2021/R-25), which provides a mandatory code of conduct for snorkellers, divers and boats involved in activities with protected species nationwide, including MRW, regardless of the MPA status [114]. This tiered approach, combining broad national conduct standards with intensive, site-specific management at high-value aggregation hubs, offers a pragmatic pathway to safeguard the biological health of the species while securing the long-term economic prosperity of the MRW industry.

Recognising the value of charismatic species as the focus for wildlife tourism is becoming increasingly widespread [2–4,11,44,115]. For example, the combined global value of mobulid meat and dried gill plates in 2013 was 100-fold less than that of the revenue generated through MRW tourism [3,116]. Although criticised as being simplistic, another method to demonstrate the disparity between consumptive and non-consumptive use of manta rays is the revenue-generating

ability of an individual animal over the duration of their lifetime [3,4,44]. *Mobula alfredi* are known to display high site fidelity [117,118] and remain within the Maldives' national borders for the duration of their lives. Less is known about *M. birostris*, but studies suggest they are less migratory than previously assumed and often have relatively limited home ranges [119]. When the current population estimate for *M. alfredi* (n = 3,500) in the Maldives is considered to be interacting with the tourism industry over their ~40-year lifespan, the population would generate approximately US$8.6 billion (assuming real discount rate of 5%; see [4]) in undiscounted cumulative gross tourism contributions. Unsurprisingly, the estimated lifetime value of an individual *M. alfredi* in the Maldives of US$2.6 million in 2021 has grown substantially in the past decade, as it was previously estimated to generate US$382,000 in its lifetime (2008; accounting for the updated life expectancy estimate of 40 years and the additional tourist expenses) [44]. These valuations can provide a relatable and quantifiable figure that can estimate the economic value of iconic species, which can highlight their value in terms of conservation through sustainable management.

## Intrinsic value of manta rays

Tour operators in the Maldives perceived manta rays to be important to the local community, emphasising their cultural significance beyond economic utility. Marine animals like whales have historically been recognised for their "natural goodness"–a concept describing the inherent value of a creature based on its ability to satisfy the necessities of its life form [120]. This recognition was a large driver of the "save the whales" movement, shifting public perception from whales as a commodity to beings with intrinsic rights to exist [120]. Our findings suggest a similar sentiment among Maldivian stakeholders; by describing manta rays through terms like "local" and "cultural significance," respondents acknowledge the rays' material goodness as a standard for evaluating human treatment of the marine environment.

This ethical foundation provides powerful grounds for conservation practices that transcend beyond financial justifications [120]. Charismatic species, such as manta rays, serve as the biological embodiment of this natural goodness, providing the motivation for conservation actions that ultimately support broader food security and climate stability [121]. As MRW tourism is reliant on the quality of the experience, it in turn promotes a healthy marine environment and engages the public and community in safeguarding the species' ability to flourish. Consequently, the conservation of charismatic species has ripple effects, yielding widespread benefits for ecosystem health by enabling the conservation of intact environments, restoration of degraded ones, conserving biodiversity, and achieving sustainable development goals [14].

Wildlife viewing tourism has the potential to provide a range of education and conservation benefits for visitors [122]. Tourists who learnt about the animals and environment from mediated wildlife watching experiences are more likely to make behaviour/lifestyle changes that benefit the environment [122]. These experiences also contribute to pro-environmental attitudes and improved on-site behaviour changes, with some tourists additionally holding longer-term intentions to engage in conservation actions [122]. An educational presentation or briefing was offered by over half of the MRW tour operators in Maldives (61%), thus reaching at least 17,000 MRW guests in 2021. Many of these tourists travelled internationally, therefore, the knowledge shared and potential behavioural changes by these tourists can be tracked outside of the Maldives, likely to Asia and Europe as tourists from India, Russia, Germany and the United Kingdom were consistently the top visitors to the Maldives [49]. For future studies, we recommend asking an additional survey question directed towards cultural significance of the target species within the indigenous society, such as myths, legends and Folklore. Such information has the potential to demonstrate a strong cultural connection and in turn support conservation.

## Caveats

The ~ 20% response rate obtained in our study is comparable to participation levels in previous wildlife tourism assessments [38,52]. For non-responding operators, missing data were estimated via online data mining; where specific operation metrics were unavailable (e.g., number of weekly MRW dives or snorkels), we applied the mean values derived from responding operators of a similar category. While the level of uncertainty introduced by this extrapolation is unknown,

it provided a necessary pathway for national-scale estimation. Furthermore, in instances where tour operators had not recently updated their websites, the collected data likely underestimated the price of snorkelling and diving trips relative to current inflation. Consequently, we have maintained a conservative approach throughout the analysis to ensure the revenue generated by manta ray tourism is not overinflated.

The benefit transfer ratios used to calculate the tourist expenses did not account for domestic transport (i.e., sea plane, speed boat, ferry and private yacht are all key modes of transport within the Maldives), souvenirs and other purchases; dissimilarly to other studies (see [3,8,52]). Some of the economic benefits of the MRW tourism industry are not easily quantified and were not considered in this study. This includes the ripple effect that tourism businesses generate in the local economy, known as economic multipliers, from purchasing goods and services, as well as from employed staff who in turn spend their salaries [52,123,124]. This study also did not account for international travel, despite the majority of visitors travelling from overseas (a large proportion traveling long distances) and therefore incurring international travel expenses [49]. Thus, it is likely that MRW tourism in the Maldives also has an impact on the global economy. Furthermore, our data collection took place in 2021, the year following the COVID-19 global pandemic, when tourist arrival in the Maldives (1.3 million) were 29% below pre-pandemic 2019 levels (1.7 million) [49]. Since then, arrivals have considerably rebounded, reaching 2.05 million in 2024 – a 55% increase from 2021 [125]. This upward trend in tourist arrivals suggests that a current economic valuation of the MRW industry would likely yield even higher figures, but it also amplifies concerns regarding the industry's carrying capacity and potential environmental pressures.

To calculate the lifetime values (*LV*) of both species, we did not apply a discount rate to account for the time value of money, whereby revenue generated in the future is typically considered less valuable than revenue realised in the present [126]. We also did not incorporate the opportunity cost of natural capital, defined as the alternative economic uses foregone by conserving manta rays for tourism rather than exploiting them for other purposes [127]. As a result, the lifetime values reported here should be interpreted as undiscounted cumulative gross tourism contributions per individual over an assumed 40-year biological lifespan [90], expressed in constant dollars. These values do not represent net present values or comprehensive economic valuations of manta rays as natural capital. While this approach may overstate the theoretical net present value, it remains a robust and pragmatic metric for conservation advocacy, effectively illustrating the scale of long-term economic loss associated with the removal of a single individual from the population [3,44].

## Conclusions

Marine wildlife tourism can be a powerful tool for both economic growth and environmental conservation. This study provides an updated assessment of MRW in the Maldives, demonstrating substantial decadal growth in the industry's scale and the flow-on benefits to the community, government and businesses. To maintain the economic integrity of the MRW tourism industry and to avoid ecological degradation and loss of income, effective management strategies must be implemented nationwide. However, a disparity remains between the amount of economic value generated through nature-based tourism and the government expenditure allocated to environmental conservation in the Maldives [97]. It is essential that national authorities recognise these benefits and ensure sufficient revenue is reinvested into protecting these iconic animals and the habitats they rely on.

The growth in MRW tourism highlights the necessity for updated regional and global assessments to support the case for conservation. By demonstrating the immense economic and social value of these animals, such valuations promote non-consumptive resource use and aid the development of region-specific management. In the face of global population declines [67,71,95,128–132], this study suggests that manta rays fulfill the necessary criteria to be classified as a flagship species for the Maldives. Their high intrinsic, social and economic importance–coupled with their cultural significance to local communities–positions them as an ideal biological anchor for broader conservation efforts. Protecting such species

not only conserves biodiversity but also fosters community engagement in conservation efforts, leveraging their cultural and economic value to motivate local stewardship. MRW tourism has the potential to continue developing into a long-term profitable and sustainable industry, resulting in sustainable livelihoods and business revenue for future generations in the years to come.

## Supporting information

**S1 Table. Data collection: details collected and sources, with shorthand for equation in data analysis.** Adapted from O'Malley et al. [3].
(DOCX)

**S1 Appendix. Tour operator surveys.** Survey questions sent to active land-based operators (i.e., activity centres in resorts and on community islands) and one for boat-based operators (i.e., liveaboards).
(DOCX)

**S2 Table. Definitions and key terms used throughout this study.**
(DOCX)

## Acknowledgments

The authors would like to offer an extended thank you to the tour operators in the Maldives who completed the survey and for those involved in the pilot stages. Many tour operator employees show incredible commitment to conserving the marine environment in the Maldives. Thank you to the *Environmental Protection Agency* staff and the *Maldives Manta Conservation Programme* staff who distributed the survey in-person to tour operators, especially Nashwa Ahmed Manik. We would like to extend our gratitude to Mary O'Malley whose guidance in the initial stages provided direction and resources for this study. Thank you to the reviewers for their constructive comments and very helpful revisions, which significantly improved the quality of this manuscript.

## Author contributions

Conceptualization: Hannah M. Moloney, Guy M.W. Stevens.

Data curation: Hannah M. Moloney, Nina Rothe, Kirsty Ballard, Florence Barraud, Farah Hamdan, Enas Mohamed Riyad, Tamaryn J. Sawers.

Formal analysis: Hannah M. Moloney, Maria I. Garcia Rojas.

Investigation: Hannah M. Moloney.

Methodology: Hannah M. Moloney, Maria I. Garcia Rojas, Guy M.W. Stevens.

Project administration: Hannah M. Moloney, Nina Rothe.

Resources: Hannah M. Moloney, Florence Barraud, Enas Mohamed Riyad.

Supervision: Asia O. Armstrong, Anthony J. Richardson, Kathy A. Townsend, Guy M.W. Stevens.

Validation: Nina Rothe, Farah Hamdan.

Visualization: Hannah M. Moloney, Maria I. Garcia Rojas.

Writing – original draft: Hannah M. Moloney, Maria I. Garcia Rojas, Nina Rothe.

Writing – review & editing: Hannah M. Moloney, Maria I. Garcia Rojas, Nina Rothe, Asia O. Armstrong, Kirsty Ballard, Florence Barraud, Farah Hamdan, Anthony J. Richardson, Enas Mohamed Riyad, Tamaryn J. Sawers, Kathy A. Townsend, Guy M.W. Stevens.

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
