## [Decision Letter · Decision Letter 0]

5 Dec 2025

PONE-D-25-29750Valuing conservation and natural wealth: The blue economy of manta ray watching in the MaldivesPLOS One

Dear Dr. Moloney,

Thank you for submitting your manuscript to PLOS ONE. After careful consideration, we feel that it has merit but does not fully meet PLOS ONE’s publication criteria as it currently stands. Therefore, we invite you to submit a revised version of the manuscript that addresses the points raised during the review process.

We look forward to receiving your revised manuscript.

Kind regards,

Joel Harrison Gayford

Academic Editor

PLOS One

Journal Requirements:

4. We note that Figure 1 in your submission contain map images which may be copyrighted. All PLOS content is published under the Creative Commons Attribution License (CC BY 4.0), which means that the manuscript, images, and Supporting Information files will be freely available online, and any third party is permitted to access, download, copy, distribute, and use these materials in any way, even commercially, with proper attribution. For these reasons, we cannot publish previously copyrighted maps or satellite images created using proprietary data, such as Google software (Google Maps, Street View, and Earth). For more information, see our copyright guidelines: http://journals.plos.org/plosone/s/licenses-and-copyright.

1. You may seek permission from the original copyright holder of Figure 1 to publish the content specifically under the CC BY 4.0 license.;

5. We note that there is identifying data in the Supporting Information file < S1 Appendix.docx>. Due to the inclusion of these potentially identifying data, we have removed this file from your file inventory. Prior to sharing human research participant data, authors should consult with an ethics committee to ensure data are shared in accordance with participant consent and all applicable local laws.

-Location data

6. If the reviewer comments include a recommendation to cite specific previously published works, please review and evaluate these publications to determine whether they are relevant and should be cited. There is no requirement to cite these works unless the editor has indicated otherwise.;

Additional Editor Comments:

You will see that both reviewers generally have a favourable view of the manuscript, but that some changes are required. Please take care to address all of the points raised by the reviewers when making your revision.;

Reviewer's Responses to Questions

**Comments to the Author**

1. Is the manuscript technically sound, and do the data support the conclusions?

Reviewer #1: No

Reviewer #2: Yes

2. Has the statistical analysis been performed appropriately and rigorously? 

Reviewer #1: No

Reviewer #2: Yes

3. Have the authors made all data underlying the findings in their manuscript fully available?

Reviewer #1: No

Reviewer #2: Yes

4. Is the manuscript presented in an intelligible fashion and written in standard English?

Reviewer #1: Yes

Reviewer #2: Yes

5. Review Comments to the Author

Reviewer #1: Review of PONE-D-25-29750

This article is an important contribution to the literature on the economic value of nature-based tourism. It serves as an important piece of information to highlight often hidden benefits of biodiversity conservation for local economies. There are significant shortcomings related to the methods used that need to be addressed before considering publication.

1. Is the manuscript technically sound, and do the data support the conclusions?

Due to methodological issues, the manuscript is not technically sound and does not support the conclusion in its current state. The manuscript requires major revisions in the methods to be considered for publication.

2. Has the statistical analysis been performed rigorously and appropriately?

• The term “total economic value” or total economic benefit” is not appropriate. Check Randall 1987. You are only looking at direct use of the use value category.

• Operator surveys: some questions are worded as if the questions were targeting the visitors, not the operators. “What was the cost per guest dive …?” is vague. Since it’s an operator survey, I assume you asked about operating costs, what the dives cost the operator, not cost the diver/customer. If so, why did you not specify subcategories to then more precisely model these expenditures in the wider economy? In any case, revenue = dives * price, not dives * cost.

• Given this issue, how sure are you that operators actually responded with their operating costs and not price charged, or vice versa? I think what you really were after was “price”, not “cost”, in order to calculate revenue. E.g. L239: In this sentence it looks like you used the term “cost” as meaning the price the operator charged for a package/experience/dive, etc.

• For non-respondents, how did you determine the number of dives these non-responding operators had in the study year? On their websites, did you look for their capacity, e.g. number of vessels or dive guides, etc. to give you a better sense than just taking the average of whatever grouping the operator fell into? How did you verify your assumptions?

• L272 – L277: What if the tour operator didn’t update the website to reflect current prices? How did you deal with outliers? Did you employ any statistical approach to identify them? Citation missing.

• With many experiences sold as packages, it’s unclear how you divided packages offered on websites into the categories you outlined in Equation 1.

• One major issue is double counting in Eq1. Tour operator revenue already includes taxes eventually paid to the government, thus you are counting them twice in Eq1, once as part of TOR and again as TX. Similar double counting occurs with guide license fees that are part of MPAR, occurring twice in Eq1, once in the TOR and again in the MPAR.

• TE is problematic with packages as opposed to paying for dives solely. Again, how did you divide up what visitors paid for packages into diving/viewing and accommodations, etc.

• Check all other equations for issues of double counting.

• Converting from Rufiya to US$ you used an exchange rate on a certain date instead of purchasing power parity that considers differences in the cost of living and is the economic approach necessary in this instance .

• Eq6 calculation of LV is not theoretically sound as it must consider the opportunity cost of natural capital, some sort of discount rate, or if omitted a discussion that justifies it.

3. Have the authors made all the data underlying the findings in their manuscript fully available?

Not yet. The authors should specify what data archive will host the data. Usually, it is better practice to publish the data in a data depository, providing the DOI to reviewers.

4. Is the manuscript presented in an intelligible fashion and written in standard English?

Well written manuscript. The following are minor suggestions to improve clarity and flow.

• Check spelling of the short title: economic or economy

• Abstract: second to last sentence, eliminate “were”. Language is a bit pretentious and inflated.

• Consider “charismatic” instead of “flagship species”. Animals are not manufactured assets, such as ships.

• Change “economic evaluation” to “economic valuation”. You are not comparing economic performance against some sort of evaluation metric or a standard but instead you quantify values, thus, use “economic valuation.” Evaluation is a different social science method.

• Many instances of the use of “this” without actually naming what this is, specifically this followed by an object. Search and replace any “this”, and either eliminate or name what “this” is. I can read that mostly you refer to the prior sentence, but your writing becomes so much more clear if you synthesize the prior sentence using a term. L 54 for example, instead of “Despite this” write: “Despite the public intent to conserve, policy makers … “

5. Additional comments:

Introduction

• Usually, intros are 2 double-spaced pages. Consider condensing significantly, perhaps to less than half the current length. Be concise on how your research fits into the broader literature. Perhaps, insert a subheading for manta ray ecology. Unless you compare the economic numbers from the literature to your results, there isn’t much need to present them all here.

• L163: unclear what negative pressure. From development or unenforced viewing regulations or some other factor.

Methods

• The first section belongs into the intro. This is background material, not methods. Instead, the methods section should start with your study design.

• What was your sample size and response rate for the pilot?

• L262, comparing response rates to other studies belongs into the discussion.

• L366: United States not “stated”

• L387-390: how did you estimate this?

• L399-404: how was this “text mining” done? A careful analysis of text is usually done following qualitative data analysis that includes determining coding categories before analyzing text. This approach can then be used to carefully assess motivations. Word clouds are problematic and should not be used in academic publications as there is no analysis, no rigor. They are good for presentations and data exploration.

Reviewer #2: This MS is well written, and the authors successfully highlight its value. First of all, as an updated study of the economic value of Maldives MRW after more than a decade is very important for future management. This study also highlights the discrepancy in staff salary revenue between Maldivian and expatriate staff, as well as the need for more stable funding that supports in-country conservation from tourism. Further, this study is also very important for backing arguments against unsustainable fishing practices (such as the recent plan to legalise shark fishing in the Maldives in 2024) and any future threats to biodiversity. Minor improvements are needed, including elaboration on fisheries and cultural significance.

6. PLOS authors have the option to publish the peer review history of their article (what does this mean?). If published, this will include your full peer review and any attached files.

Reviewer #1: No

Reviewer #2: **Yes:** Muhammad Ichsan

You may also use PLOS’s free figure tool, NAAS, to help you prepare publication quality figures: https://journals.plos.org/plosone/s/figures#loc-tools-for-figure-preparation

---

## [Author Response · Author response to Decision Letter 1]

21 Jan 2026

Letter from Editor

Date: Dec 05 2025 10:06PM

To: "Hannah Maree Moloney" hannah.moloney@mantatrust.org;hannah.moloney@research.usc.edu.au

From: "PLOS ONE" plosone@plos.org

Subject: PLOS ONE Decision: Revision required [PONE-D-25-29750]

Attachment(s): Review MI.docx

PONE-D-25-29750

Valuing conservation and natural wealth: The blue economy of manta ray watching in the Maldives

PLOS One

Dear Dr. Moloney,

Thank you for submitting your manuscript to PLOS ONE. After careful consideration, we feel that it has merit but does not fully meet PLOS ONE’s publication criteria as it currently stands. Therefore, we invite you to submit a revised version of the manuscript that addresses the points raised during the review process.

● A rebuttal letter that responds to each point raised by the academic editor and reviewer(s). You should upload this letter as a separate file labeled 'Response to Reviewers'.

● A marked-up copy of your manuscript that highlights changes made to the original version. You should upload this as a separate file labeled 'Revised Manuscript with Track Changes'.

● An unmarked version of your revised paper without tracked changes. You should upload this as a separate file labeled 'Manuscript'.

We look forward to receiving your revised manuscript.

Kind regards,

Joel Harrison Gayford

Academic Editor

PLOS One

Response: Thank you for giving us the opportunity to resubmit our manuscript. We have addressed the feedback and provided point by point detailed replies below and in the “Response to Reviewers” letter.

Journal Requirements:

Response: Changed. The manuscript now complies with PLOS ONE’s style template.

Response: Changed. Data is now available through Zenodo using DOI 10.5281/zenodo.18252206

Response: Changed. We have removed the ethics statement from the Acknowledgment section and moved it to the Methods (under Study Design, Tour operator surveys):

“The research was carried out under the Environmental Protection Agency Protected Species Research Permit (EPA/2021/PSR-M09) and in accordance with the University of the Sunshine Coast Human Ethics exemption (E24002).”

4. We note that Figure 1 in your submission contain map images which may be copyrighted. All PLOS content is published under the Creative Commons Attribution License (CC BY 4.0), which means that the manuscript, images, and Supporting Information files will be freely available online, and any third party is permitted to access, download, copy, distribute, and use these materials in any way, even commercially, with proper attribution. For these reasons, we cannot publish previously copyrighted maps or satellite images created using proprietary data, such as Google software (Google Maps, Street View, and Earth). For more information, see our copyright guidelines: http://journals.plos.org/plosone/s/licenses-and-copyright.

Response: Changed. Thank you for the suggestions. We were able to locate the reference for the spatial data we have used through the Ocean Plus website https://habitats.oceanplus.org. We have also contacted Ocean Plus via email to request additional permissions using PLOS ONE’s “Content-permission-form”, however we are yet to receive a response.

We have updated the figure caption to contain all reference material:

“Fig. 1. Geographical distribution of the 10 primary manta ray watching sites (shown by red circles) in the Republic of Maldives’. The 26 geographical atolls of the Maldives’ archipelago are shown by the red box in panel a. The capital, Malé City is identified by the yellow triangle and the nation’s 20 administrative divisions are highlighted in blue in panel b. Although Malé City is an administrative division, for this study we have grouped it with Kaafu, thus, only 20 divisions are shown. The manta ray watching sites correspond with those outlined in Table 2. This map was created in R Studio using reef features from Millennium Coral Reef Mapping Project (MCRMP; https://habitats.oceanplus.org); MCRMP validated maps provided by the Institute for Marine Remote Sensing, University of South Florida (IMaRS/USF) and Institut de Recherche pour le Développement (IRD, Centre de Nouméa), with support from NASA. IRD does not endorse these products [75].”

REFERENCE:

UUNEP-WCMC. Ocean+ Habitats [On-line]. 2020. doi:https://doi.org/10.34892/fpe3-ar97

5. We note that there is identifying data in the Supporting Information file < S1 Appendix.docx>. Due to the inclusion of these potentially identifying data, we have removed this file from your file inventory. Prior to sharing human research participant data, authors should consult with an ethics committee to ensure data are shared in accordance with participant consent and all applicable local laws.

-Location data

Response: Changed. We have removed all personal data from <S1 Appendix.docx >.

Response: Thank you for clarifying.

Additional Editor Comments:

You will see that both reviewers generally have a favourable view of the manuscript, but that some changes are required. Please take care to address all of the points raised by the reviewers when making your revision.

Response: Thank you also for your feedback and suggestions. We have taken the time to carefully address each point made by both reviewers and the editor.

Comments to the Author

1. Is the manuscript technically sound, and do the data support the conclusions?

Reviewer #1: No

Reviewer #2: Yes

2. Has the statistical analysis been performed appropriately and rigorously?

Reviewer #1: No

Reviewer #2: Yes

3. Have the authors made all data underlying the findings in their manuscript fully available?

Reviewer #1: No

Reviewer #2: Yes

Response: Changed. Data is now available through Zenodo using DOI 10.5281/zenodo.18252206

4. Is the manuscript presented in an intelligible fashion and written in standard English?

Reviewer #1: Yes

Reviewer #2: Yes

Reviewer #1

5. Review Comments to the Author

Review of PONE-D-25-29750

This article is an important contribution to the literature on the economic value of nature-based tourism. It serves as an important piece of information to highlight often hidden benefits of biodiversity conservation for local economies. There are significant shortcomings related to the methods used that need to be addressed before considering publication.

Response: Thank you for your positive feedback and helpful revisions particularly regarding the methods.

1. Is the manuscript technically sound, and do the data support the conclusions?

Due to methodological issues, the manuscript is not technically sound and does not support the conclusion in its current state. The manuscript requires major revisions in the methods to be considered for publication.

Response: Changed. We have addressed all points made during the revision process, particularly ones related to the Methods where changes were suggested.

2. Has the statistical analysis been performed rigorously and appropriately?

• The term “total economic value” or total economic benefit” is not appropriate. Check Randall 1987. You are only looking at direct use of the use value category.

Resp

---

## [Decision Letter · Decision Letter 1]

6 Mar 2026

PONE-D-25-29750R1Valuing conservation and natural wealth: The blue economy of manta ray watching in the MaldivesPLOS One

Dear Dr. Moloney,

Thank you for submitting your manuscript to PLOS ONE. After careful consideration, we feel that it has merit but does not fully meet PLOS ONE’s publication criteria as it currently stands. Therefore, we invite you to submit a revised version of the manuscript that addresses the points raised during the review process.

A letter that responds to each point raised by the academic editor and reviewer(s). You should upload this letter as a separate file labeled 'Response to Reviewers'.

We look forward to receiving your revised manuscript.

Kind regards,

Joel Harrison Gayford

Academic Editor

PLOS One

Journal Requirements:

If the reviewer comments include a recommendation to cite specific previously published works, please review and evaluate these publications to determine whether they are relevant and should be cited. There is no requirement to cite these works unless the editor has indicated otherwise.;

Reviewers' comments:

Reviewer's Responses to Questions

**Comments to the Author**

1. If the authors have adequately addressed your comments raised in a previous round of review and you feel that this manuscript is now acceptable for publication, you may indicate that here to bypass the “Comments to the Author” section, enter your conflict of interest statement in the “Confidential to Editor” section, and submit your "Accept" recommendation.

Reviewer #1: All comments have been addressed

Reviewer #2: All comments have been addressed

2. Is the manuscript technically sound, and do the data support the conclusions?

Reviewer #1: Yes

Reviewer #2: Yes

3. Has the statistical analysis been performed appropriately and rigorously? 

Reviewer #1: Yes

Reviewer #2: Yes

4. Have the authors made all data underlying the findings in their manuscript fully available?

Reviewer #1: Yes

Reviewer #2: Yes

5. Is the manuscript presented in an intelligible fashion and written in standard English?

Reviewer #1: Yes

Reviewer #2: Yes

6. Review Comments to the Author

Reviewer #1: I thank the authors for their continued and quite extensive efforts to improve this manuscript. I can accept the revisions and the paper should be published after some additional minor modifications, that I may suggest as follows:

• Figures are all low dpi, increase dpi to at least 300dpi, Fig 3 fonts need to be legible, Fig. 4 has missing axis and missing labels … what are you showing?

• Lines 574-577. Explain the different methods used in the past and that you use and then give an honest assessment of your estimate describing factors contributing to over and underestimating the true value. It’s good that you cite numbers to validate growth but your estimate needs more discussion how it fits past estimates and how not. Also, check the statement in the quoted lines. To me it is a bit contradictory. Consider revising to eliminate the contradiction.

The discussion section could benefit from a few suggested points:

- Discrepancy between foreign and Maldivian staff salaries, despite the latter most likely having better local knowledge. You could specifically point out the distribution of benefits between local and foreign staff with policy implications. Why are foreign guides needed? Could the government levy a tax for foreign guides and use that income to bolster local guides’ workforce development, education, etc. Also, note that the involvement of locals in conservation-oriented tourism such as manta ray guiding and promoting local employment in this sector with rising wages could significantly play into competing successfully against opportunities locals have in the illegal harvesting sector of the manta ray economy. Harvesters may have traditional knowledge that they could immediately deploy in the tourism sector if wages are better there.

- Hanifaru seems to have much higher ray density. Why is that? Are protections working? If so, could those be applied to other areas to increase ray habitat and populations there and perhaps distribute visitor pressure across more viewing locations? Your numbers speak to implementing the same rules that Hanifaru has to other areas. Are any of the visitation rates limited by regulation, managing congestion and negative impact on ray health?

Reviewer #2: All my comments have been satisfactorily addressed by the authors.

Publications on manta ray tourism values in the Maldives are mostly non-species-specific, and some targeting manta rays are outdated, making this publication an original and updated source of information.

The authors acquired and stated the ethics permit and permit exemption required and ensured the anonymity of respondents in the MS.

7. PLOS authors have the option to publish the peer review history of their article (what does this mean?). If published, this will include your full peer review and any attached files.

Reviewer #1: No

Reviewer #2: No

---

## [Author Response · Author response to Decision Letter 2]

24 Mar 2026

Reviewer Comments to the Author

Reviewer #1: I thank the authors for their continued and quite extensive efforts to improve this manuscript. I can accept the revisions and the paper should be published after some additional minor modifications, that I may suggest as follows:

Response: Thank you for your positive feedback, as well as your helpful revisions which have improved the manuscript.

• Figures are all low dpi, increase dpi to at least 300dpi, Fig 3 fonts need to be legible, Fig. 4 has missing axis and missing labels … what are you showing?

Response: Changed. All figures are 300 dpi, the font size on Fig 3 is now larger making it more legible, and Fig 4 now has labels (key and north arrow). We have not added axis labels as this figure is more of an infographic than a graph, however we agree that the contents of Fig 4 needed to be made clearer from the stand-alone figure. All files have been re-uploaded with increased resolution during the resubmission process.

Fig 4. The lifetime value of individual manta rays (both species) as part of tourism calculated using the combined economic revenue from manta ray watching in the Republic of Maldives in 2021. Tourism that focused on Mobula alfredi (reef manta rays) was identified in 18 administrative regions, compared to only one region for M. birostris (oceanic manta rays; Gnaviyani/Fuvahmulah). The geographical extent of manta ray tourism for each species is shown using coloured dotted rectangles.

• Lines 574-577. Explain the different methods used in the past and that you use and then give an honest assessment of your estimate describing factors contributing to over and underestimating the true value. It’s good that you cite numbers to validate growth but your estimate needs more discussion how it fits past estimates and how not. Also, check the statement in the quoted lines. To me it is a bit contradictory. Consider revising to eliminate the contradiction.

Response: Changed. Thank you for this suggestion. We have revised the opening paragraph of the Discussion to provide a transparent comparison of methodologies and a balanced assessment of our valuation. The statement that was in lines 574-577 has been revised and eliminates the contradiction.

Discussion:

“Economic benefits of manta ray watching tourism

We demonstrate that marine wildlife tourism provides immense value to local and national economies by generating revenue, supporting cultural heritage, and creating employment opportunities. Using the Maldives as a case study, this study provides an updated national valuation of the economic benefits of MRW tourism. We found that the 475,000 dive and snorkel trips taken by MRW guests in 2021 generated an estimated US$227.3 million, comprising US$39 million in revenue for tour operators and US$188.3 million in revenue of secondary tourist expenditures. These estimates represent a substantial increase from previous assessments in 1997 (US$7.8 million) and 2008 (US$8.1 million) [43,44]. When the industry was last valued at a national scale in 2011 (using 2008 data), it was estimated to generate US$15.5 million in combined revenue, including US$7.4 million in tourist expenditures derived from the 157,000 MRW trips [3].

While these estimates suggest significant expansion, direct comparisons are complicated by evolving methodologies. The 2011 study [3] utilised a site-density approach for MRW sites, whereas our study employed a high-resolution approach combining tour operator surveys and online data mining. Both estimates are inherently conservative, accounting only for quantifiable metrics; however, the 2011 estimate was limited by its focus on tourism density at known MRW sites, and the benefit transfer ratio used to calculate secondary tourist expenses did not account for pricing tiers between operator types (e.g., expenses at a luxury resorts vs at a community island) [3]. Further, our 2021 estimate may be influenced by broader market inflation and a more comprehensive capture of secondary supply chains that were previously unmonitored.

Despite these methodological differences, the 2021 value reflects a remarkable >380% growth in tour operator revenue over the past decade. The growth underscores the increasing economic significance of MRW tourism and mirrors the trajectory of the broader Maldivian tourism sector, which reported a 286% increase in GDP contribution over the same period (2008 = US$593.1 million and 2021 = US$2.3 billion) [49]. Consequently, while the different methods utilised over the last two decades prevent precise direct comparisons of absolute values, the overarching trend clearly establishes MRW tourism as a cornerstone of the national economy.”

The discussion section could benefit from a few suggested points:

- Discrepancy between foreign and Maldivian staff salaries, despite the latter most likely having better local knowledge. You could specifically point out the distribution of benefits between local and foreign staff with policy implications. Why are foreign guides needed? Could the government levy a tax for foreign guides and use that income to bolster local guides’ workforce development, education, etc. Also, note that the involvement of locals in conservation-oriented tourism such as manta ray guiding and promoting local employment in this sector with rising wages could significantly play into competing successfully against opportunities locals have in the illegal harvesting sector of the manta ray economy. Harvesters may have traditional knowledge that they could immediately deploy in the tourism sector if wages are better there.

Response: Changed. We have updated the employment section in the Discussion to incorporate these constructive suggestions regarding the socio-economic distribution of tourism benefits. While manta rays are not harvested in the Maldives, we argue that failing to provide competitive career trajectories for locals may result in the loss of invaluable human capital, specifically Traditional Ecological Knowledge.

Regarding the suggestion to discuss specific government levies or the reasons for foreign guide recruitment; we have respectfully omitted a detailed analysis of these policy mechanisms. While foreign guides are commonly hired to cover specific skill gaps or service requirements (such as specialised language skills or academic background in marine biology), we feel that a deep dive into labour-market structures or specific recruitment requirements is beyond the scope of this study’s primary economic focus. We have instead focused the discussion on the broader economic necessity of upskilling and equitable benefit retention to support long-term conservation.

Discussion:

“Both the Maldivian economy and the livelihood of its people are largely dependent on marine resources [97]. This study provides evidence for the flow-on socio-economic impacts of manta ray tourism in the Maldives. Almost 4,000 staff were employed by tour operators to work directly on MRW diving and snorkelling tourism activities, generating US$27 million per year in staff salary revenue. Despite the 2,602 Maldivian staff representing 67% of the MRW tourism workforce, they only received 55% of the distributed staff salary within the industry. A similar discrepancy was observed in reef shark-diving tourism in 2016, where Maldivians represented 55% of the workforce but received 50% of the distributed staff salary [52]. The wage gap likely stems from the disproportionate employment of Maldivian nationals in lower-paid operational roles, such as divemaster, equipment maintenance, or boat operations, while expatriates more frequently occupy senior management and instructional positions, a pattern observed in many Global South tourism industries [98,99].

While some operators in the Maldives are already implementing initiatives to further address these disparities, encouraging the widespread adoption of a structured approach toward localisation remains vital. Beyond fostering individual career progression, these initiatives offer broader benefits, including enhanced local economic development, long-term economic sustainability, and reduced recruitment costs [98,100]. Formalising and rewarding the high-value traditional ecological knowledge possessed by Maldivian guides with competitive, senior-level wages is essential for ensuring that marine tourism offers a lucrative and prestigious career path. Maldivians often possess knowledge regarding manta ray aggregations and oceanographic patterns that is indispensable for high-quality guest experiences. If the tourism sector fails to provide a career trajectory that competes with other high-paying industries, this human capital may be lost to sectors that do not contribute to the blue economy or conservation. By incentivising the promotion of locals into specialist roles, the Maldives can ensure that the economic benefits of its natural wealth are not only generated but also equitably retained within its communities [98,100].”

- Hanifaru seems to have much higher ray density. Why is that? Are protections working? If so, could those be applied to other areas to increase ray habitat and populations there and perhaps distribute visitor pressure across more viewing locations? Your numbers speak to implementing the same rules that Hanifaru has to other areas. Are any of the visitation rates limited by regulation, managing congestion and negative impact on ray health?

Response: Changed. We have updated the Management and sanctuaries section in the Discussion to clarify that while Hanifaru’s manta ray density is driven by unique conditions, its management success provides a vital proof-of-concept for the rest of the Maldives.

Discussion:

“Management and sanctuaries

There has never been a targeted commercial fishery for manta rays in the Maldives; however, these populations are still vulnerable to bycatch, illegal fishing, habitat degradation, and unregulated tourism [36,69,70]. The Maldivian government’s recognition of the economic value of iconic animals such as manta rays, whale sharks, and reef sharks, was instrumental in developing legal frameworks and policies such as the Hanifaru MPA, and the protection of shark and ray species [35,36]. Importantly, successful conservation relies on community support; a lack of buy-in and top-down conservation approaches can lead to non-compliance or policy reversal [105]. Recent attempts to reestablish longline fisheries and legalise shark fishing in the Maldives in 2024 threatened the sanctuary status, but were successfully blocked through community pushback and an online poll [106,107]. Yet in 2025, the government reopened the gulper shark (Centrophorus spp.) fishery, an action met with complex attitudes and a strong community opposition [108]. These events underscore the need for robust conservation strategies that align with national conservation goals with international commitments to the United Nations Convention on Biological Diversity (CBD) and the Conservation of Migratory Species of Wild Animals (CMS).

Nationwide there are 42 MPAs in the Maldives covering 116.3 km2 (0.5% of the 21,596 km2 of Economic Exclusive Zone), yet for many years Hanifaru remained the only MPA with active, site-specific monitoring and enforcement [36,58,69]. The high density and predictability of manta rays at Hanifaru MPA (mean of 32 individuals per trip) is driven by the geomorphological and oceanographic conditions, eddies that trap high concentrations of their zooplankton prey, making it a globally unique aggregation [109]. Although such densities of manta rays infrequently occur elsewhere in the Maldives, the Hanifaru MPA model demonstrates how rigorous regulation can successfully mitigate the impacts of high visitor pressure [69]. Since its designation in 2009, regulations have limited congestion by capping the number of visitors at 80 snorkellers and five boats at any one time, prohibiting SCUBA diving, enforcing a code-of-conduct and requiring licensed guides [69,110]. In 2012, the entirety of Baa Atoll was formally designated as the Baa Atoll Biosphere Reserve, with Hanifaru MPA as a core protected zone. Revenue generated through the Biosphere Reserve supports the Baa Atoll Conservation Fund, which is utilised for community education and research [110].

In 2025, drawing from the success of Hanifaru MPA, long-term monitoring and stakeholder consultation led to the introduction of active monitoring and enforcement of the South Ari Marine Protected Area (SAMPA), a critical aggregation area for whale sharks [111-113]. By implementing regulations and revenue-generating mechanisms modelled after Hanifaru MPA at other key megafauna aggregation sites, the government could prevent the degradation of high-traffic habitats while maintaining high-quality tourism experiences. A significant step towards this national scaling occurred in 2023 with the development of the Protected Species Regulation (2021/R-25), which provides a mandatory code of conduct for snorkellers, divers and boats involved in activities with protected species nationwide, including MRW, regardless of the MPA status [114]. This tiered approach, combining broad national conduct standards with intensive, site-specific management at high-value aggregation hubs, offers a pragmatic pathway to safeguard the biological health of the species while securing the long-term economic prosperity of the MRW industry.”

Reviewer #2: All my comments have been satisfactorily addressed by the authors.

Publications on manta ray tourism values in the Maldives are mostly non-species-specific, and some targeting manta rays are outdated, making this publication an original and updated source of information.

The authors acquired and stated the ethics permit and permit exemption required and ensured the anonymity of respondents in the MS.

Response: Thank you for your positive and helpful feedback.

---

## [Editor Report · Decision Letter 2]

13 Apr 2026

Valuing conservation and natural wealth: The blue economy of manta ray watching in the Maldives

PONE-D-25-29750R2

Dear Dr. Moloney,

We’re pleased to inform you that your manuscript has been judged scientifically suitable for publication and will be formally accepted for publication once it meets all outstanding technical requirements.

Kind regards,

Joel Harrison Gayford

Academic Editor

PLOS One
---

## [Editor Report · Acceptance letter]

PONE-D-25-29750R2

PLOS One

Dear Dr. Moloney,

I'm pleased to inform you that your manuscript has been deemed suitable for publication in PLOS One. Congratulations! Your manuscript is now being handed over to our production team.

Kind regards,

on behalf of

Mr. Joel Harrison Gayford

Academic Editor

PLOS One